# SARS-CoV-2 Spike triggers barrier dysfunction and vascular leak via integrins and TGF-β signaling

Scott B. Biering [1,12] ✉, Francielle Tramontini Gomes de Sousa[1,12], Laurentia V. Tjang [1], Felix Pahmeier [1], Chi Zhu[2,3], Richard Ruan[1], Sophie F. Blanc [1], Trisha S. Patel[1], Caroline M. Worthington[4], Dustin R. Glasner[5,6], Bryan Castillo-Rojas[1], Venice Servellita[5,6], Nicholas T. N. Lo [1], Marcus P. Wong[1], Colin M. Warnes [1], Daniel R. Sandoval[7], Thomas Mandel Clausen[7], Yale A. Santos[5,6], Douglas M. Fox [1,8], Victoria Ortega[9], Anders M. Näär [2,3], Ralph S. Baric [10], Sarah A. Stanley[1,8], Hector C. Aguilar [9], Jeffrey D. Esko[7], Charles Y. Chiu [3,5,6,11], John E. Pak[4], P. Robert Beatty[1,8] & Eva Harris [1,8] ✉

Severe COVID-19 is associated with epithelial and endothelial barrier dysfunction within the lung as well as in distal organs. While it is appreciated that an exaggerated inflammatory response is associated with barrier dysfunction, the triggers of vascular leak are unclear. Here, we report that cell-intrinsic interactions between the Spike (S) glycoprotein of SARS-CoV-2 and epithelial/endothelial cells are sufficient to induce barrier dysfunction in vitro and vascular leak in vivo, independently of viral replication and the ACE2 receptor. We identify an S-triggered transcriptional response associated with extracellular matrix reorganization and TGF-β signaling. Using genetic knockouts and specific inhibitors, we demonstrate that glycosaminoglycans, integrins, and the TGF-β signaling axis are required for S-mediated barrier dysfunction. Notably, we show that SARS-CoV-2 infection caused leak in vivo, which was reduced by inhibiting integrins. Our findings offer mechanistic insight into SARS-CoV-2-triggered vascular leak, providing a starting point for development of therapies targeting COVID-19.

Severe acute respiratory syndrome coronavirus-2 (SARS-CoV-2) is a human pathogen belonging to the *Coronaviridae* family and the causative agent of coronavirus disease 2019 (COVID-19). Outcomes of SARS-CoV-2 infection range from asymptomatic to non-severe COVID-

19 with flu-like symptoms that can progress to severe cases associated with acute lung injury and acute respiratory distress syndrome (ARDS)[1–3]. The lung pathology of severe COVID-19 involves pulmonary edema stemming from epithelial and endothelial barrier dysfunction

[1]Division of Infectious Diseases and Vaccinology, School of Public Health, University of California, Berkeley, Berkeley, CA, USA. [2]Department of Nutritional Sciences and Toxicology, University of California, Berkeley, Berkeley, CA, USA. [3]Innovative Genomics Institute, University of California, Berkeley, Berkeley, CA, USA. [4]Chan Zuckerberg Biohub, San Francisco, CA, USA. [5]Department of Laboratory Medicine, University of California, San Francisco, San Francisco, CA, USA. [6]UCSF-Abbott Viral Diagnostics and Discovery Center, San Francisco, CA, USA. [7]Department of Cellular and Molecular Medicine, Glycobiology Research and Training Center, University of California, San Diego, La Jolla, CA, USA. [8]Department of Molecular and Cell Biology, University of California, Berkeley, Berkeley, CA, USA. [9]Department of Microbiology and Immunology, Cornell University, Ithaca, NY, USA. [10]Department of Epidemiology, University of North Carolina at Chapel Hill, Chapel Hill, NC, USA. [11]Department of Medicine, University of California, San Francisco, San Francisco, CA, USA. [12]These authors contributed equally: Scott B. Biering, Francielle Tramontini Gomes de Sousa. ✉e-mail: sbiering@berkeley.edu; eharris@berkeley.edu

believed to be induced by an exacerbated inflammatory response;[4–6] however, the specific triggers of epithelial/endothelial hyperpermeability and involvement of particular viral factors are not well understood.

SARS-CoV-2 possesses a ~30-kb positive-sense RNA genome encoding ~29 proteins including four structural proteins: spike (S), nucleocapsid (N), matrix (M), and envelope (E)[7,8]. Homotrimers of the S glycoprotein coat the SARS-CoV-2 virion and engage the viral receptor, angiotensin converting enzyme 2 (ACE2), on the surface of susceptible cells to mediate viral entry[9,10]. S consists of two subunits, S1−containing the receptor-binding domain (RBD) that engages ACE2, and S2− containing the fusion machinery required for virus-cell membrane fusion[7,11,12]. Two cleavage sites, S1/S2 and S2′, separate S1 and S2 and must be cleaved by host proteases for S to mediate virus-cell fusion. Furin-like proteases, cathepsin L, and TMPRSS2 are able to cleave these sites, making them essential host factors for SARS-CoV-2 infection[10,13–15]. RBD engagement of ACE2 triggers conformational changes in S that result in S1 shedding and insertion of the fusion peptide into the host membrane[16,17].

In addition to ACE2, the SARS-CoV-2 S glycoprotein has been reported to engage numerous cell-surface factors, including heparan sulfate-containing proteoglycans (HSPG) and integrins, which are proposed to serve as attachment factors promoting SARS-CoV-2 entry[18–20]. Beyond facilitating viral entry, the engagement of S with these host factors may mediate signaling pathways contributing to lung pathology. Indeed, it was demonstrated that engagement of ACE2 by SARS-CoV-1 S results in depletion of ACE2 from the cell surface, leading to an imbalance in the renin-angiotensin system and thus promoting inflammatory responses, barrier dysfunction, and lung injury[21–23]. A comparable ACE2-dependent pathway has been described for SARS-CoV-2 S[24–28]. A unique element of the SARS-CoV-2 entry cascade is that the RBD-containing S1 portion of S can be shed from the surface of virions following engagement of the ACE2 receptor, suggesting that shed-S1 may also interact with epithelial and endothelial cells independently of the virion[16,17]. While interactions between the SARS-CoV-2 S glycoprotein and the cell surface may promote barrier dysfunction, the mechanisms by which this occurs and the host factors involved are not well understood.

We and others have described a phenomenon by which viral proteins, such as the flavivirus non-structural protein 1 (NS1), interact with endothelial cells to trigger signaling cascades that mediate disruption of cellular structures required for endothelial barrier integrity, including the endothelial glycocalyx layer (EGL) and intercellular junctional complexes[29–34]. Here, we investigated whether SARS-CoV-2 S contributes to endothelial and epithelial barrier dysfunction in vitro and vascular leak in vivo. Our study reveals that full-length S and the RBD from SARS-CoV-2 are sufficient to mediate barrier dysfunction and vascular leak in an ACE2-independent manner. Further, transcriptional analyses showed that S modulates expression of transcripts involved in regulation of the extracellular matrix (ECM), and experimental validation revealed a mechanism in which glycosaminoglycans (GAGs), integrins, and transforming growth factor beta (TGF-β) signaling are required for barrier dysfunction. Finally, we find that SARS-CoV-2 infection in vivo triggers vascular leak in the lungs of mice, which is reversed by antagonizing integrins. These data uncover the role of SARS-CoV-2 S in promoting barrier dysfunction and provide critical mechanistic insight into this process.

## Results

### SARS-CoV-2 S mediates endothelial and epithelial hyperpermeability

To determine whether SARS-CoV-2 S can mediate barrier dysfunction independently from SARS-CoV-2 infection, we utilized a trans-epithelial/endothelial electrical resistance (TEER) assay. Using TEER, we measured the electrical resistance across a monolayer of human pulmonary microvascular endothelial cells (HPMEC) or human lung epithelial cells (Calu-3) seeded in the apical chamber of Transwells as a proxy for barrier permeability (Fig. 1A). We selected HPMECs as a representative endothelial cell because of the lung pathology associated with COVID-19 and because they do not endogenously express ACE2 and are not permissive to SARS-CoV-2 infection, thus allowing us to separate viral infection from S-mediated barrier dysfunction (Fig. S1A, B). We produced recombinant soluble trimeric S and RBD in-house, which we determined were of high purity (Fig. S1C, D, G).

Parental HPMEC were treated with soluble trimeric SARS-CoV-2 S, as well as dengue virus (DENV) NS1 and vascular endothelial growth factor (VEGF) as positive controls. As seen previously, treatment of HPMEC with DENV NS1 led to a reversible decrease in barrier resistance that peaked around 6 hours post-treatment (hpt) (Fig. 1B). Interestingly, SARS-CoV-2 S also triggered a reversible barrier dysfunction in parental HPMEC but with distinct temporal kinetics, reaching a peak ~24 hpt. Since HPMEC do not endogenously express ACE2, these data suggest that S triggers barrier dysfunction in a manner independently of ACE2 (Fig. 1B). Given that S-triggered endothelial hyperpermeability peaked at 24 hpt, we utilized this timepoint for subsequent experiments. We next treated both parental HPMEC or HPMEC overexpressing human ACE2 (HPMEC/ACE2) with soluble trimeric SARS-CoV-2 S and found that S triggered endothelial hyperpermeability in a dose-dependent manner (Fig. 1C). We found that S triggered barrier dysfunction comparably in both parental HPMEC and HPMEC/ACE2, further suggesting that this phenotype is independent of ACE2 (Fig. 1C). To determine whether virion-bound S was also able to trigger endothelial hyperpermeability, we utilized vesicular stomatitis virus (VSV) pseudotyped with SARS-CoV-2 S (VSV-S), along with VSV expressing VSV glycoprotein (VSV-G) or no glycoprotein (VSV-bald) as negative controls. Comparable to soluble S and the VEGF positive control, VSV-S also triggered endothelial hyperpermeability in a dose-dependent manner, whereas VSV-G and VSV-bald had no effect on barrier function. Again, no significant differences were observed in relative TEER values when comparing results for HPMECs and HPMEC/ACE2 (Fig. 1D). Further, we found that recombinant SARS-CoV-2 S RBD was sufficient to trigger a drop in TEER in HPMEC in a dose-dependent manner (Fig. 1E). We next tested the capacity of S to trigger hyperpermeability of epithelial Calu-3 cells in a TEER assay and found that full-length trimeric S and the RBD were indeed sufficient to induce epithelial hyperpermeability (Fig. 1F). To confirm that the phenotype was specific to S, we measured the ability of anti-S antibodies to inhibit S-induced barrier dysfunction and found that a cocktail of two anti-S antibodies abolished S-mediated endothelial hyperpermeability (Fig. 1G). To determine if the capacity of SARS-CoV-2 S to trigger endothelial hyperpermeability in HPMEC was conserved among related coronaviruses, we generated full-length S and RBD from human coronaviruses (HCoV)−229E and HCoV-OC43, which we determined to be of high purity (Fig. S1E−G). Interestingly, we found that S and RBD from HCoV-229E and HCoV-OC43 were unable to trigger endothelial hyperpermeability in HPMEC, in contrast to SARS-CoV-2 S, suggesting this property is particular to SARS-CoV-2 S (Fig. S1H). Taken together, these data indicate that SARS-CoV-2 S and its RBD can facilitate barrier hyperpermeability in both human lung endothelial and epithelial cells.

### SARS-CoV-2 S triggers disruption of the endothelial and epithelial glycocalyx layer

Cell surface proteins on epithelial and endothelial surfaces are surrounded by a dense mesh of glycans termed the epithelial/endothelial glycocalyx layer (EGL). The EGL includes sialic acid and glycosaminoglycans (GAGs) and serves as a critical determinant of barrier function, protecting epithelial and endothelial cells from shear stress[35]. Our previous work has demonstrated that flavivirus NS1 glycoproteins mediate endothelial dysfunction through disruption of the EGL via activation of EGL-degrading enzymes[30,32]. To determine if SARS-CoV-2

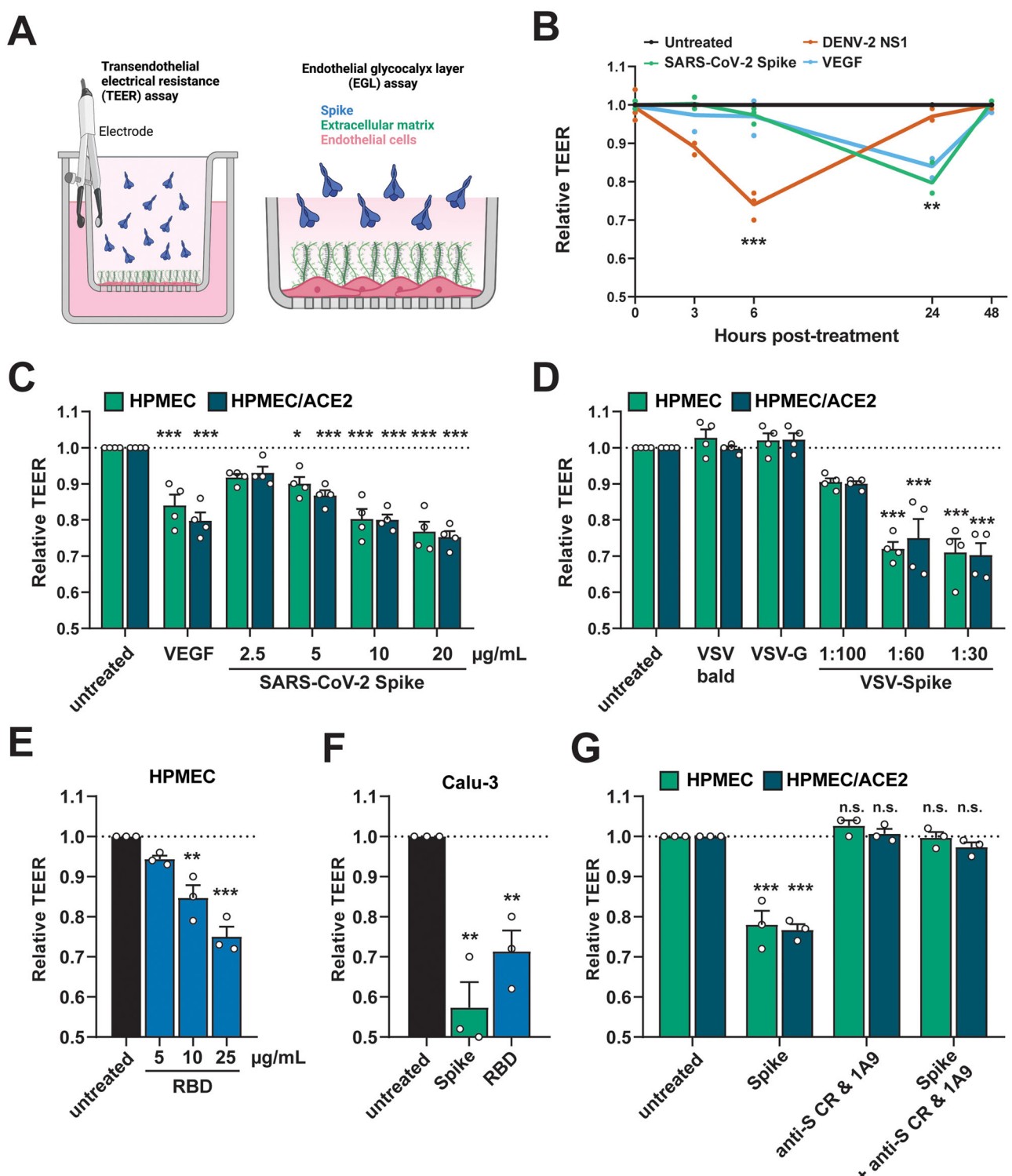

S disrupts the EGL, we treated HPMEC and Calu-3 cells with S and measured the surface levels of key EGL components, including sialic acid (SIA), heparan sulfate (HS), hyaluronic acid (HA), and chondroitin sulfate (CS) by immunofluorescence assay (IFA). We also measured the levels of hyaluronidase and neuraminidase, which degrade hyaluronan and sialylated glycans, respectively. We found that S treatment of both HPMEC and Calu-3 cells resulted in a significant decrease in EGL components compared to control conditions, and conversely, led to upregulation of EGL-degrading enzymes (Fig. 2A–H). These data suggest that, like flavivirus NS1, SARS-CoV-2 S mediates disruption of the EGL.

Although our parental HPMEC are not permissive to SARS-CoV-2 infection and had no detectable levels of ACE2 as determined by Western blot, we utilized CRISPR-Cas9 to introduce double-stranded breaks in the ACE2 gene to rule out the possibility that low levels of ACE2 present on the cell surface could be contributing to our phenotype. We found that S-mediated EGL disruption was comparable in HPMECs transduced with lentivirus encoding ACE2-targeting guide RNAs or non-targeting guides (NT), further supporting that SARS-CoV-2 S-triggered barrier dysfunction can occur independently of ACE2 (Fig. S2A, B). Finally, we tested whether S from multiple SARS-CoV-2 variants of concern could trigger barrier dysfunction and found that,

**Fig. 1 | SARS-CoV-2 S triggers endothelial and epithelial barrier hyperpermeability. A** Schematic depicting S-triggered barrier dysfunction measured by a trans-endothelial/epithelial electrical resistance assay (TEER; left) and an endo/epithelial glycocalyx layer (EGL) assay (right). **B** Time course of TEER assay measuring the barrier function of HPMEC monolayers over time with the indicated treatments, including DENV2 NS1 (5 μg/mL), VEGF (50 ng/mL), and SARS-CoV-2 S (10 μg/mL). Data are from $n = 3$ biological replicates. **C** A TEER assay measuring the barrier of monolayers of HPMEC and HPMEC/ACE2 at 24 hours after the indicated treatments. VEGF positive control (50 ng/mL). Data are from $n = 4$ biological replicates. **D** Same as **C** but treated with the indicated VSV-pseudotyped particles at the indicated dilutions. VSV-bald and VSV-G are diluted 1:30. Data are from $n = 4$ biological replicates. **E** Same as **C** but treated with the indicated concentrations of SARS-CoV-2 RBD. Data are from $n = 3$ biological replicates. **F** Same as **C** but

measuring the barrier of Calu-3 cell monolayers. Data are from $n = 3$ biological replicates. **G** A TEER inhibition assay measuring the capacity of a cocktail of anti-S antibodies to inhibit S-mediated endothelial hyperpermeability. S (10 μg/mL) and the antibody cocktail (15 μg/mL of each antibody; 1A9 [Genetex] and CR3022 [Absolute Antibody]) were added simultaneously to the upper chamber of Transwell inserts to a monolayer of HPMEC or HPMEC/ACE2, and TEER was measured 24 hours post-treatment (hpt). Data are from $n = 3$ biological replicates. In all panels, the dotted line is the normalized TEER value of the untreated control condition. All data are plotted as mean +/− SEM. For all panels, values are compared to untreated controls by One-Way ANOVA with Tukey's Multiple comparisons test except for (**B**), which was analyzed by two-sided unpaired $t$-test. $*p < 0.05$, $**p < 0.01$, $***p < 0.001$, and n.s. $p > 0.05$. Source data are provided as a Source Data file.

comparable to S protein from the ancestral Wuhan variant, S derived from alpha, beta, gamma, delta, and omicron variants could all facilitate EGL disruption on HPMEC, indicating that this phenomenon is not specific to one variant, but instead is a generalizable property of SARS-CoV-2 S (Fig. S2C, D). Finally, to determine whether SARS-CoV-2 virions could facilitate EGL disruption, we inoculated HPMEC with S or with SARS-CoV-2 live virus at an MOI of 5. We found that both recombinant S and viral particles were sufficient to trigger EGL disruption, providing additional evidence that both free S and virion-associated S could trigger endothelial dysfunction (Fig. S2E, F).

### SARS-CoV-2 S and viral infection trigger vascular leak in vivo

To determine whether SARS-CoV-2 S could mediate vascular leak in vivo, we utilized our previously characterized dermal leak model, which involves local intradermal injection of compounds into distinct spots in the dorsal dermis of mice followed by intravenous administration of dextran conjugated to Alexa fluor 680 to serve as a tracer. Following a 2-hour treatment, mice were euthanized and the local relative accumulation of dextran-680 in the dorsal dermis was measured by a fluorescent scanner[31,33,36]. Given our observations that SARS-CoV-2 S mediates barrier dysfunction in an ACE2-independent manner, we utilized WT C57BL/6 J mice that do not express human ACE2 and are not permissive to infection by most SARS-CoV-2 variants, including the Wuhan and Washington isolates[9,37]. We observed that, comparably to the DENV NS1 positive control, SARS-CoV-2 S induced vascular leak in the dorsal dermis of mice above control conditions, indicating that SARS-CoV-2 S is sufficient to trigger vascular leak in vivo (Fig. 3A, B). To test if SARS-CoV-2 S could mediate vascular leak when administered in a more physiologically relevant route, we administered S intranasally and then measured accumulation of dextran-680 in various organs to evaluate both local (lungs) and distal (spleen, small intestine, liver, and brain) vascular leak. We found that SARS-CoV-2 S significantly induced vascular leak locally in the lungs as well as distally in the spleen and small intestine, with trending but non-significant leak measured in the liver and brain, as determined through accumulation of dextran-680 (Figs. 3C–H and S2A–D).

To determine if SARS-CoV-2 infection triggers vascular leak in vivo, we infected K18-hACE2 transgenic mice with the WA/1 human isolate of SARS-CoV-2. Mice were sacrificed at the peak of disease, and sections of mouse lungs and small intestine were fixed and subjected to hematoxylin and eosin (H&E) staining. We observed a significant influx of inflammatory cells into the lungs and small intestine of SARS-CoV-2-infected mice compared to control mice. Further, we observed dispersal of red blood cells throughout lungs and small intestines in infected mice relative to control mice (Figs. 3I, J, S3E, F). To examine SARS-CoV-2 vascular leak in vivo in a quantitative manner, we measured accumulation of dextran-680 in the lungs of C57BL/6 J mice infected with a mouse-adapted strain of SARS-CoV-2 (MA10). At the peak of disease at 7 days post-infection, we administered the fluorescence tracer dextran-680 intravenously and evaluated fluorescence accumulation in lungs of mice. We found a significant accumulation of

fluorescence signal in infected mice relative to control mice, which positively correlated with the viral inoculum (Fig. 3K, L). Taken together, these observations show that SARS-CoV-2 infection induces barrier dysfunction and vascular leak in mice.

### Glycosaminoglycans are required for S-mediated barrier dysfunction

Given that S interactions with HSPGs enhance infectivity of SARS-CoV-2[20,38], we hypothesized that cell surface GAGs are also required for S-mediated barrier dysfunction. We measured the capacity of heparin, a highly charged linear polysaccharide analogous to heparan sulfate, to antagonize S-mediated endothelial hyperpermeability and found that heparin decreased S-induced endothelial hyperpermeability in a TEER assay, supporting our hypothesis that interactions with sulfated glycans on the cell surface contribute to S-mediated barrier dysfunction (Fig. 4A). To further test the contributions of glycans to S-mediated barrier dysfunction, we utilized recombinant enzymes to remove specific glycans from the surface of endothelial cells, including HS (heparin lyases), HA (hyaluronidase), CS (chondroitinase), or sialic acid (neuraminidase), and tested the ability of S to trigger barrier dysfunction under each of these conditions. We found that removal of HS, HA, or CS, but not sialic acid, inhibited S-mediated endothelial hyperpermeability, further supporting a role for GAGs in S-mediated barrier dysfunction (Figs. 4B and S4A) as measured by TEER. Removal of HS and HA was also sufficient to abrogate S-mediated EGL disruption, as measured by cell surface levels of sialic acid (Fig. 4C, D). To genetically investigate the contribution of HS to this phenotype, we utilized CRISPR-Cas9 technology to produce cell lines with two genes individually knocked out that are involved in the HS proteoglycan biosynthetic pathway; namely, SLC35B2 and XYLT2[38–41]. We confirmed that these KO cell lines possessed less detectable HS on the cell surface as compared to the NT controls (Fig. S4B, C). When we treated these cell lines with S, we found that they were less susceptible to S-mediated barrier dysfunction compared to the NT controls, providing further evidence of the involvement of GAGs in this pathway (Fig. 4E, F).

### Extracellular matrix modulating-components contribute to SARS-CoV-2 S-mediated barrier dysfunction

We next hypothesized that components reported to be critical for flavivirus NS1-mediated EGL disruption may contribute to S-mediated EGL disruption. Therefore, we investigated the involvement of several endogenous enzymes and factors known to regulate the homeostasis of the ECM, including cathepsin L, heparanase (HPSE), A Disintegrin And Metalloprotease 17 (ADAM17), IL-6R, and matrix metalloproteinase 9 (MMP9). Cathepsin L, HPSE, and MMP9 are required for flavivirus NS1-mediated endothelial barrier dysfunction[30,31,42,43]. MMP9 is reported to be involved in release of the ECM-modulating TGF-β[44]. ADAM17 has been implicated in modulation of COVID-19 pathogenesis through regulation of ACE2 shedding, IL-6 signaling, and TGF-β

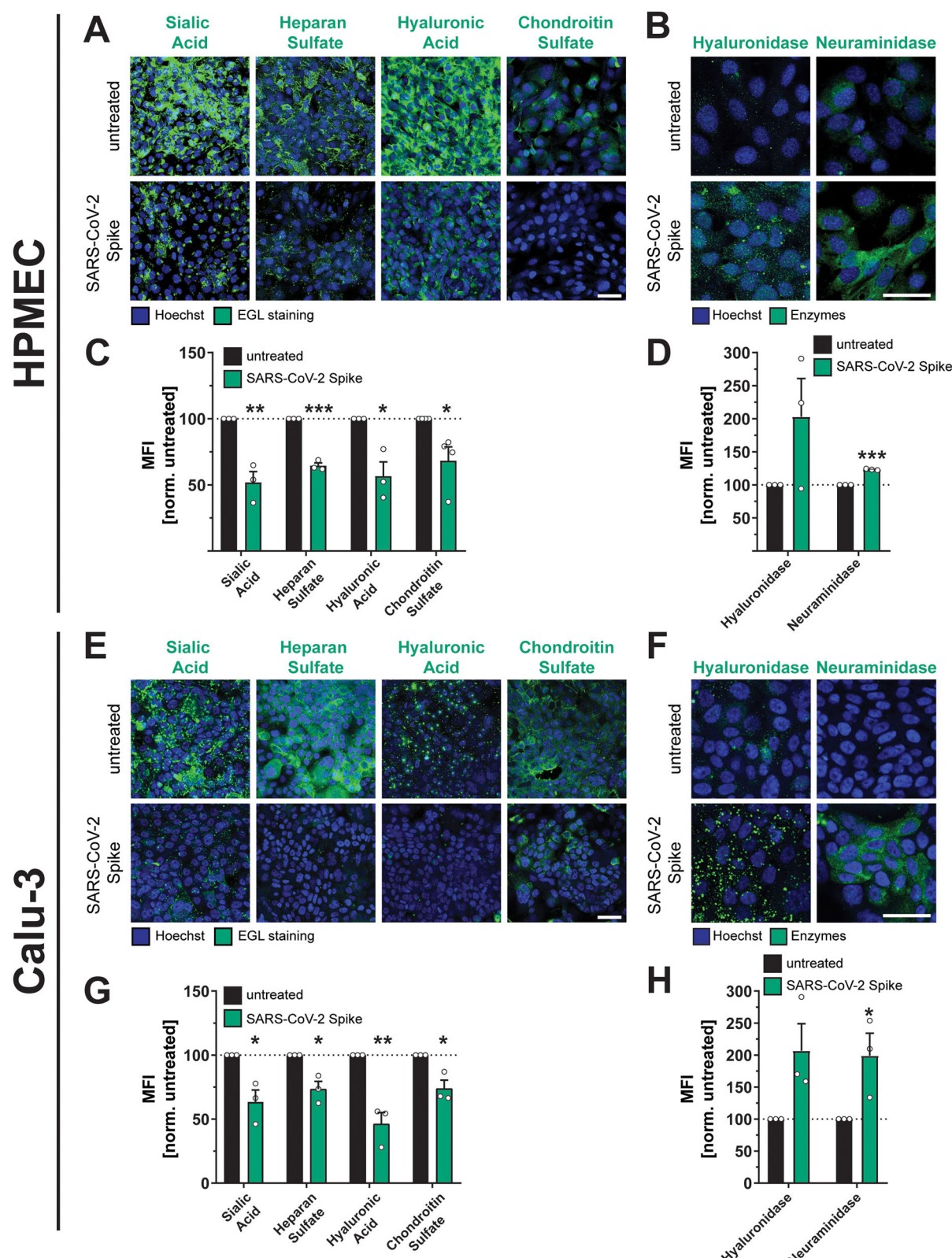

signaling[45,46]. Using CRISPR/Cas9, we produced KO HPMEC for each of these factors and found that HPMECs genetically deficient in HPSE, ADAM17, and MMP9 were no longer sensitive to S-mediated barrier dysfunction. In contrast, HPMECs deficient for cathepsin L and IL-6R displayed comparable barrier dysfunction to NT control HPMEC (Figs. 4G and S4D–H). These data highlight the requirement for critical modulators of the EGL for S-mediated barrier dysfunction and further

reveal differences by which S and flavivirus NS1 mediate barrier dysfunction.

**Transcriptional analysis reveals SARS-CoV-2 S modulation of genes involved in extracellular matrix homeostasis**
To gain further insight into the effects of S on endothelial cells, we conducted RNA sequencing (RNA-seq) to measure the global

**Fig. 2 | SARS-CoV-2 S facilitates disruption of the endothelial and epithelial glycocalyx layer. A** An immunofluorescence microscopy-based EGL disruption assay measuring levels of the indicated glycans on the surface of HPMECs. After 24 h of SARS-CoV-2 S (10 μg/mL) treatment, cells were fixed and then stained without permeabilization. Displayed are representative images from *n* = 3 biological replicates. **B** Same as **A**, but cells were permeabilized before staining for the indicated EGL disrupting enzymes. Displayed are representative images from *n* = 3 biological replicates. **C** Quantification of **A** from *n* = 3 biological replicates. **D** Quantification of **B** from *n* = 3 biological replicates. **E** Same as **A** but measuring EGL disruption of Calu-3 cell monolayers. Displayed are representative images from *n* = 3 biological replicates. **F** Same as **B** but measuring expression of EGL-disrupting enzymes in Calu-3 cells. Displayed are representative images from *n* = 3 biological replicates. **G** Quantification of **E** from *n* = 3 biological replicates. **H** Quantification of **F** from *n* = 3 biological replicates. For all images, nuclei were probed with Hoechst in blue and the indicated glycans in green with scale bars at 50 μm. Dotted lines are the normalized untreated control conditions. MFI is mean fluorescence intensity. All data are plotted as mean + /− SEM with *\*p* < 0.05, *\*\*p* < 0.01, *\*\*\*p* < 0.001, and n.s. *p* > 0.05 by two-sided unpaired *t*-test compared to untreated controls. Source data are provided as a Source Data file.

transcriptional profile of HPMEC and HPMEC/ACE2 following treatment with S. In S-treated HPMEC, we identified 65 differentially expressed genes (DEGs) compared to untreated controls, including 45 upregulated and 20 downregulated genes; in S-treated HPMEC/ACE2, there were 42 DEGs, with 34 upregulated and 7 downregulated compared to untreated controls. Further, the DEGs obtained from HPMEC and HPMEC/ACE2 were similar, indicating ACE2 expression has minimal impact on the HPMEC transcriptional response to S (Figs. 5A, B, S5A, B, Tables S1, S2). We observed numerous genes encoding ECM components, including proteoglycans, collagens and integrins, as well as genes coding for proteins involved in ECM degradation, such as *CAPN2* (calpain), an endothelial cysteine protease, and *HTRA1*, which is responsible for degradation of fibronectin and proteoglycans, in agreement with our observations that S disrupts the EGL[47]. Upregulation of *XYLT2* (xylosyltransferase 2), encoding an enzyme responsible for biosynthesis of CS, HS, and dermatan sulfate proteoglycans, is potentially involved in an EGL recovery pathway, post-S-mediated barrier dysfunction. Transcriptome profiling also showed activation of the TGF-β signaling pathway, demonstrated by upregulation of *TGFBI*, *LTBP2*, *LTBP3*, *SERPINE1*, *FBLN5*, *POSTN*, *FN1*, *THBS1*, *BGN*, *ITGB5*, and *ITGA4* genes. TGF-β is a well-known mediator of cellular differentiation, proliferation, and migration, with an important role in the regulation of vascular permeability and inflammatory responses[48]. We next conducted a protein-protein interaction network analysis of upregulated genes from parental HPMEC treated with S to examine the relationships between these DEGs. We found that S-upregulated genes form a highly interconnected matrix with predicted protein interactions, pinpointing 31 genes of ECM or ECM-associated proteins, including numerous components of the focal adhesion complex that connects the cytoskeleton of barrier cells to the ECM via integrins[49] (Fig. 5C). Given these observations, in addition to a requirement for glycans, we hypothesized that S-mediated barrier dysfunction requires integrins, which in turn promote TGF-β maturation and signaling.

## Integrins are required for S-mediated barrier dysfunction

Integrins are transmembrane proteins that are critical for maintaining barrier function by connecting cells to the ECM[50]. Integrins interact with factors regulating ECM homeostasis, such as the latency associated peptide (LAP), which non-covalently binds to TGF-β, maintaining it in an inactive form[51]. LAP releases the active form of TGF-β in response to diverse stimuli, including mechanical stress, or through competition with integrin-binding factors (resulting in the release of LAP from the ECM)[51]. Integrins mediate these interactions via a small peptide motif (RGD) on interacting proteins[51,52]. Recombinant RGD peptides are commonly used as integrin-binding motifs that can compete for integrin binding with other RGD-containing proteins like LAP and TGF-β[53]. Further, it has been reported that SARS-CoV-2 S has evolved an RGD motif within its RBD[19]. However, although SARS-CoV-2 S has been shown to bind to integrins, the functional relevance of the RGD motif is unclear[18]. To test for a role of integrins in S-mediated barrier dysfunction, we measured the capacity of an integrin inhibitor (ATN-161) to antagonize S-mediated barrier dysfunction in both a TEER assay and an EGL disruption assay. We found that ATN-161 treatment of cells abrogated S-mediated barrier dysfunction relative to control

conditions in a dose-dependent manner, while having no effect alone on barrier function (Figs. 6A, B, S6A, B). The dependence of S-mediated barrier dysfunction on integrins was independent of ACE2 expression, as S-mediated TEER reduction was equivalently abrogated by ATN-161 in both HPMEC and HPMEC/ACE2 cells (Fig. 6A). To confirm the role of integrins in S-mediated endothelial hyperpermeability in vivo, we tested the capacity of ATN-161 to inhibit leak in our intradermal mouse model. We found that ATN-161 significantly inhibited S-mediated vascular leak while not inducing leak by itself, relative to a PBS control (Fig. 6C, D). We next tested whether a recombinant RGD peptide, mimicking the RGD motif within the SARS-CoV-2 S RBD, was sufficient to mediate endothelial hyperpermeability and EGL disruption in HPMEC. We utilized a KGD peptide (mimicking the corresponding sequence in SARS-CoV-1 S) as well as a DRG scrambled peptide as negative controls. Intriguingly, we found that RGD was sufficient to trigger endothelial hyperpermeability and EGL disruption comparably to S, whereas the effect of KGD and DRG on barrier function was comparable to the untreated control conditions (Figs. 6E, F, and S6C). Next, we tested the ability of the RGD peptide to trigger vascular leak in vivo in the intradermal leak model and found that, in agreement with the corresponding in vitro data, the RGD peptide was sufficient to induce vascular leak in a dose-dependent manner and at comparable levels to full-length S (Fig. 6G, H). To genetically confirm a role for integrins, we used CRISPR-Cas9 to knock out two RGD-binding integrins shown to interact with S, integrin alpha-5 (α5) and integrin beta-1 (β1) (Fig. 6I). We found that S-mediated barrier dysfunction was significantly inhibited in both KO HPMEC lines relative to the NT control HPMEC, with the caveat that relative barrier function was lower, compared to NT controls, in ITGA5 KO HPMECS (Figs. 6J, K, and S6D). To test for a role for integrins in SARS-CoV-2 infection-triggered vascular leak in vivo, we again infected C57BL/6 mice with SARS-CoV-2 MA10 and administered the integrin inhibitor ATN-161 daily throughout the course of infection. We then administered dextran-680 and evaluated leak in mice at day 7 post-infection. We found, as before, that infection of mice triggered significant leak in the lungs relative to mock-infected conditions. Notably, while daily administration of ATN-161 had no effect on background levels of vascular leak, it did significantly inhibit SARS-CoV-2-triggered leak, suggesting that virus-induced vascular leak, comparably to S-triggered leak, requires integrins (Fig. 6L, M). Taken together, these data indicate that integrins play a critical role in S- and virus-mediated barrier dysfunction and that the RGD peptide is sufficient to induce hyperpermeability.

## TGF-β signaling is essential for SARS-CoV-2 S-mediated barrier dysfunction

Integrins are key regulators of TGF-β maturation and signaling[48,54]. Given the critical role of integrins for SARS-CoV-2 S-mediated barrier dysfunction as well as our RNA-Seq data revealing S-mediated TGF-β transcriptional upregulation, we hypothesized that the RGD motif of SARS-CoV-2 S engages integrins, releasing mature TGF-β, which in turn activates signaling via interactions with the TGF-β receptor (TGFBR) and results in barrier dysfunction. To test this hypothesis, we measured levels of TGF-β in the supernatant of HPMECs treated with S and observed a significant increase in TGF-β in supernatants of S-treated

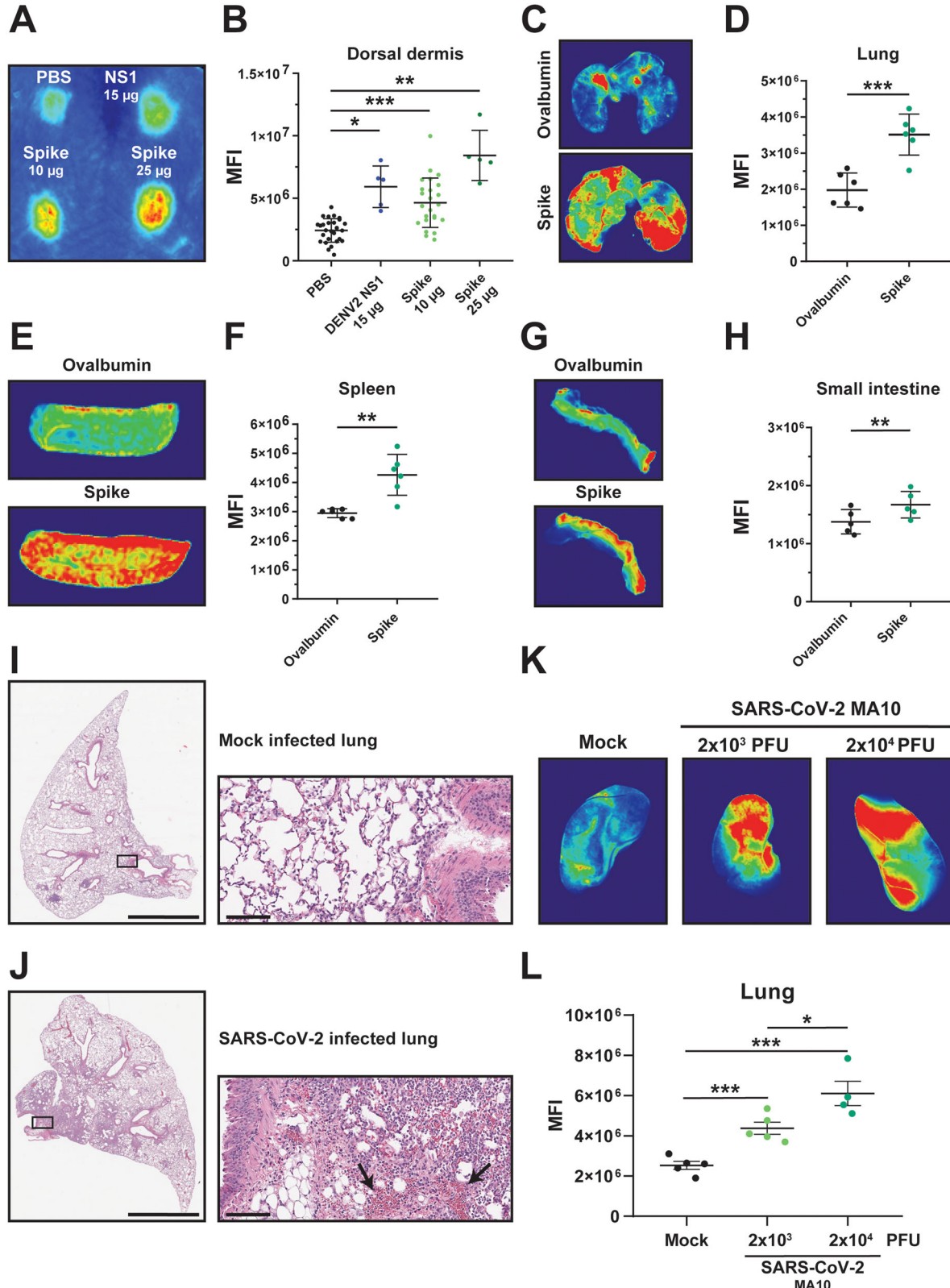

cells compared to untreated control conditions (Fig. 7A). To determine the functional consequence of this TGF-β production, we treated HPMEC with recombinant TGF-β and found that this was sufficient to trigger endothelial hyperpermeability (Fig. 7B). To determine the contribution of TGF-β signaling to S-mediated endothelial dysfunction, we antagonized TGF-β signaling via antibody blockade of TGFBR1 and found that cells treated with anti-TGFBR1 were less sensitive to

S-mediated endothelial hyperpermeability compared to IgG isotype control conditions (Figs. 7C and S7A). Importantly, the anti-TGFBR1 antibody did not trigger endothelial hyperpermeability on its own (Fig. S7A). We next utilized a small-molecule inhibitor of TGF-β signaling (SB431542) and found that cells and mice treated with this molecule were less responsive to S-mediated TEER reduction, EGL disruption and vascular leak compared to the vehicle controls

**Fig. 3 | SARS-CoV-2 S and SARS-CoV-2 infection triggers vascular leak in vivo.**
**A** A representative mouse back from a dermal leak experiment. The dorsal dermises of mice were injected intradermally with the treatments and doses indicated. Mice then received a dextran-680 tracer molecule intravenously. Mouse dermises were collected 2 h post-treatment, and quantification of local dermal leak was assessed by a fluorescent scanner. **B** Quantification of **A** from mice treated with PBS ($n = 27$), DENV2 NS1 (15 μg; $n = 5$), S (10 μg; $n = 25$), and S (25 μg; $n = 5$). **C** Representative lung images from a SARS-CoV-2 S systemic vascular leak assay. Mice were administered 50 μg of SARS-CoV-2 S or ovalbumin intranasally as indicated, and 22 hpt were administered a dextran-680 tracer intravenously as in **A**. Organs of mice were collected 2 hours post dextran-680 administration (24 hours post-S treatment), and accumulation of dextran-680 was measured with a fluorescent scanner. **D** Quantification of **C** from $n = 6$ mice. **E** Same as **C** except representative images of spleens are shown. **F** Quantification of **E** from $n = 6$ mice. **G** Same as **C** except representative images of small intestine are shown. **H** Quantification of **G** from $n = 5$

mice. **I, J** Hematoxylin and eosin (H&E) staining was performed on lung sections from K18-hACE2 mice 7 days post-infection with 100 TCID$_{50}$ units of SARS-CoV-2 WA/1 isolate. Displayed are representative images of lungs from $n = 3$ mice in mock-infected conditions (**I**) and from $n = 4$ mice infected with SARS-CoV-2 (**J**); left panels with scale bars at 2 mm and right panels consisting of zoomed-in insets with scale bars at 100 μm. Arrows point to dispersed red blood cells. **K** Representative lung images from C57BL/6 mice infected with the indicated dose of SARS-CoV-2 mouse-adapted strain (MA-10) for 7 days. Mice were administered a dextran-680 tracer intravenously as in **A**. Lungs were collected 2 hours after dextran-680 administration and fixed overnight in formalin, and the fluorescence accumulation was measured with a fluorescent scanner. **L** Quantification of **K** from $n = 5$ mice, except for the $2 \times 10^4$ PFU condition, which was from $n = 4$ mice. MFI is mean fluorescence intensity. All data are plotted as mean $+/-$ SEM with $*p < 0.05$, $**p < 0.01$, and $***p < 0.001$ by two-sided unpaired $t$-test. Source data are provided as a Source Data file.

(Figs. 7D–G, S7B, C). Finally, we utilized CRISPR-Cas9 to produce TGFBR1 KO HPMECs (Fig. 7H). We found that these KO cells were less sensitive to S-mediated barrier dysfunction relative to the NT control cells (Figs. 7I, J, and S7D). Taken together, these data demonstrate that SARS-CoV-2 S triggers enhanced release of TGF-β from HPMEC and highlight a critical role for TGF-β signaling in S-mediated barrier dysfunction.

## Discussion

Our study reveals the capacity and mechanism by which SARS-CoV-2 S mediates barrier dysfunction in epithelial and endothelial cells in vitro and vascular leak in vivo, thus suggesting that S alone can mediate barrier dysfunction independently from viral infection[55]. Our work indicates that levels of S observed in clinical samples from COVID-19 patients are sufficient to mediate barrier dysfunction (2.5 μg/ml)[56]. Our findings suggest that, in addition to functioning in viral entry, S interactions with GAGs and integrins induce vascular leak via activation of the TGF-β pathway[18,57]. Further, our study offers a mechanistic explanation for the overproduction of TGF-β during COVID-19, which has been correlated with disease severity[58,59]. Thus, our work defines a new mechanism for S-triggered barrier dysfunction and vascular leak and uncovers potential new therapeutic avenues for COVID-19 treatment.

The levels of S used in our study ranged from 2.5-20 μg/mL, with the majority of experiments conducted at 10 μg/mL (equivalent to ~50 nM). These concentrations are in line with levels detected circulating in patients as well as extrapolated from viral loads detected in sputum from critically ill COVID-19 patients (ranging from ng/mL to μg/mL levels)[56,60–64]. We also hypothesize that local concentrations of S accumulating in capillaries deep within tissues would likely be higher than levels circulating in patient sera. Thus, the concentrations of S we utilized in our study are consistent with circulating levels found in severe COVID-19 patients. However, the source of S that interacts with endothelial and epithelial cells to mediate barrier dysfunction during SARS-CoV-2 infection is still unclear. Our data suggest that virion-associated full-length S, soluble trimeric S, and recombinant RBD of S are sufficient to trigger barrier dysfunction. Thus, we propose that SARS-CoV-2 can trigger barrier dysfunction through multiple avenues, including (1) during infection of virus-permissive cells, (2) through shedding of soluble S1 after enzymatic cleavage following ACE2 interactions on a cell, (3) through expression of S on the surface of infected cells that can interact with neighboring cells, and (4) through interactions with ACE2-negative non-permissive cells. Further investigation of clinical samples as well as in vivo experiments are required to explore these possibilities.

Based on our genetic data defining host factors required for S-mediated barrier dysfunction, we propose a model by which SARS-CoV-2 S first engages GAGs on the cell surface via positively charged surfaces in the RBD acquired specifically by SARS-CoV-2[20]. Once bound to the cell surface, S can engage integrins, such as α5β1, via an RGD

integrin-binding motif within the RBD[18]. Engagement of integrins displaces LAP, which under steady-state conditions maintains TGF-β in an inactive state, thus resulting in the release of mature TGF-β that in turn engages the TGFBR to mediate signaling pathways regulating transient barrier dysfunction (Fig. 7K). This compromise of barrier function is likely a result of activation of key enzymes such as HPSE, hyaluronidases, neuraminidases, MMP9, and ADAM17, which have distinct roles in disruption of the EGL and intercellular junctional complexes. Further, MMP9 and ADAM17 have separate reported roles in mediating maturation of TGF-β; thus, they may also contribute to S-mediated barrier dysfunction through this process[44,45]. This pathway is supported by the observations of others reporting a role for heparan sulfate in S cell binding, integrins in S-mediated endothelial cell activation and barrier dysfunction, and TGF-β as a correlate of COVID-19 disease severity[20,28,58,59]. While this investigation begins to shed light on the mechanisms by which SARS-CoV-2 S triggers barrier disruption, further studies are required to define additional host factors as well as to determine the relative contribution of each factor to this pathway.

Our study uncovers a new potential role of S beyond ACE2 binding and viral entry, and many critical questions remain. First and foremost is how S-mediated barrier dysfunction influences outcome of SARS-CoV-2 infection. Previous data indicate that NS1-mediated vascular leak can exacerbate a sublethal DENV infection, providing direct evidence that soluble NS1 can promote DENV pathogenesis[29,65]. In addition to inflammatory responses directly triggered by DENV NS1[65], one hypothesis by which NS1 promotes pathogenesis is through facilitating dissemination of blood-borne flaviviruses from the blood into distal tissues where the virus can replicate to high titers[32,42,55,66]. Thus, we speculate that a potential contribution of S-mediated barrier dysfunction to COVID-19 pathogenesis is to promote dissemination of SARS-CoV-2 from the lung to the blood, and then into distal organs where virus-permissive cells reside. This is exemplified by the observation that administration of S into the lungs of mice results in systemic leak in the spleen and small intestine (Fig. 3). This could help explain the diverse clinical manifestations observed in COVID-19 patients, and in fact persistent circulation of S has been described in patients experiencing post-acute COVID-19 sequelae[61]. Further, the surface of the lungs is covered with a dense glycocalyx comprising many proteoglycans and glycoproteins, with a primary constituent being mucus composed of membrane-tethered and gel-forming mucins. It has been recently demonstrated that these mucins aid cells to be refractory to SARS-CoV-2 infection due to steric hindrance of virus-cell interactions; thus, S-mediated barrier dysfunction disrupting the EGL may make virus-permissive epithelial cells more accessible to invading virus[67,68].

Our observation that S from SARS-CoV-2 but not from HCoV-229E or HCoV-OC43 triggers endothelial hyperpermeability of HPMECs suggests that the capacity to trigger barrier dysfunction in these lung cells is not conserved equivalently among all coronaviruses. We

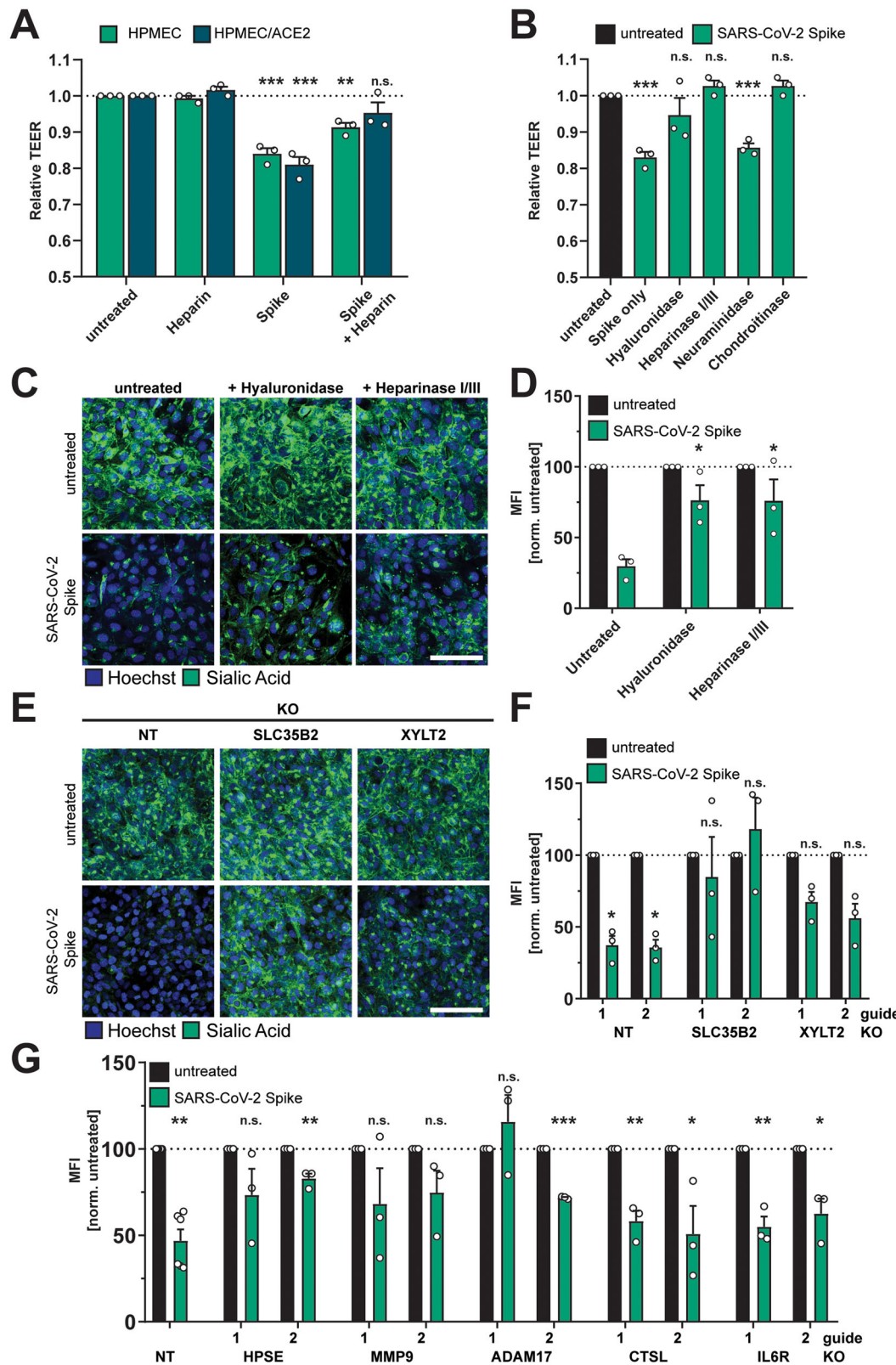

hypothesize that the increased capacity of SARS-CoV-2 S to interact with heparan sulfate and integrins on HPMEC may explain this specificity, but additional studies are required to test this possibility[18,20]. Further, the expression of ACE2 on the cell surface may influence the capacity of S proteins to interact with endothelial and epithelial cells and trigger barrier dysfunction. This may be the case for SARS-CoV-1 and HCoV-NL63, which both utilize ACE2 as an entry receptor[69,70]. The

interaction of S from both SARS-CoV-1 and HCoV-NL63 with Vero-E6 cells has been shown to lead to downregulation of ACE2 expression, although via a different mechanism, which has been shown to contribute to tissue injury in the case of SARS-CoV-1 S[21,23,71]. Importantly, several reports have demonstrated that SARS-CoV-2 S can also trigger inflammatory responses and perturb barrier function in an ACE2-dependent manner[24,25,27,72,73]. It will be critical to understand the relative

**Fig. 4 | Glycosaminoglycans and EGL-modulating enzymes are required for SARS-CoV-2 S-mediated barrier dysfunction. A** TEER inhibition assay on monolayers of HPMECs or HPMEC/ACE2 treated with heparin (10 μg/mL), S (10 μg/mL), or both simultaneously. TEER readings were taken 24 hpt. Data are from $n = 3$ biological replicates. **B** A TEER inhibition assay where monolayers of HPMEC were treated with recombinant hyaluronidase (10 μg/mL), heparin lyases I and III (5 mU/mL each), neuraminidase (1 U/mL), or chondroitinase (25 mU/mL) simultaneously with S (10 μg/mL) treatment. TEER readings were taken 24 hpt. Data are from $n = 3$ biological replicates. **C** EGL inhibition assay on HPMEC treated with hyaluronidase (10 μg/mL) or heparin lyases I and III (5 mU/mL each) and simultaneously treated with S (10 μg/mL) and fixed 24 hpt. **D** Quantification of **C** from $n = 3$ biological replicates. Statistics are comparisons of indicated conditions to the S-only control condition. **E** Representative images from an EGL disruption assay of HPMEC transduced with lentiviruses encoding the indicated gene-targeting guide RNA.

Cells were treated with 10 μg/mL S, and sialic acid was visualized by IFA 24 hpt. **F** Quantification of **E** from $n = 3$ biological replicates. Control guide data from this panel are from the same experiment as Fig. S2B. **G** Same as **E** and **F** but using the indicated guide RNAs. Non-target (NT) data are pooled from two cell line replicates. Control guide data from this panel are from the same experiments as Fig. 7J. For all panels, sialic acid is stained with Wheat Germ Agglutinin in green and nuclei are stained with Hoechst in blue with scale bars at 50 μm. MFI is mean fluorescence intensity. Dotted lines are the normalized untreated control conditions. All data are plotted as mean +/− SEM with $*p < 0.05$, $**p < 0.01$, $***p < 0.001$, and n.s. $p > 0.05$ by One-Way ANOVA with Tukey's Multiple comparisons test except for (**G**) which was analyzed by two-sided unpaired $t$-test. Statistics in panels **A**, **B**, **F**, and **G** are comparisons to untreated controls and in panel **D** are comparisons to the S-only control condition. Source data are provided as a Source Data file.

contribution of the ACE2-independent vs. ACE2-dependent pathways to vascular leak in vivo and define which pathways a given coronavirus S protein can trigger.

Our observation that S is sufficient to mediate endothelial dysfunction and vascular leak allows a direct comparison with the flavivirus NS1 protein. Such a comparison contributes to our understanding of how viruses activate signaling pathways to mediate barrier dysfunction and leads to the concept of development of pan anti-leak therapeutics targeting multiple soluble viral proteins. Both similarities and distinctions are apparent in the mechanisms by which S and NS1 mediate endothelial barrier dysfunction[55]. Our mechanistic investigation uncovered the contributions of GAGs (HS, CS, HA), MMP9, ADAM17, HPSE, integrins, and TGF-β signaling to S-mediated barrier dysfunction. GAG binding and activation of enzymes like MMP9, HPSE, hyaluronidase, and neuraminidases are common requirements for both NS1- and S-mediated endothelial dysfunction. In contrast, cathepsin L appears to be essential only for NS1 pathogenesis[30,31], while S-mediated dysfunction requires engagement of integrins and TGF-β signaling. These differences may explain the distinct kinetics of our in vitro hyperpermeability assays, with NS1 causing a peak of barrier dysfunction ~6 hours post-NS1 treatment while the S-mediated peak occurs ~24 hpt with no perturbation of barrier function observed at 6 hpt. One potential reason for this may be due to the requirement for TGF-β production and signaling as a second messenger for S-mediated endothelial dysfunction, which is not required for flavivirus NS1-induced leak. Further comparative investigations between the mechanisms of S- and NS1-mediated barrier dysfunction are needed to fully understand what makes these pathways similar yet distinct.

Although integrins and TGF-β are required for S-mediated vascular leak, their role in modulating SARS-CoV-2 infection in vivo is undoubtedly complex. For example, the well-characterized roles of integrins and TGF-β in immune cell adhesion/extravasation, modulation of inflammatory responses, and tissue repair will likely have complex effects on SARS-CoV-2 viral infection in humans[74,75]. Further, in addition to the pathogenic consequences of vascular leak in COVID-19, barrier dysfunction in the lung may also promote infiltration of immune cells that can clear virus, which would be predicted to be beneficial to the host; however, overactivation of these immune cells can lead to the "cytokine storm" typically associated with ARDS in severe COVID-19. Dissecting the differential effects of vascular leak on SARS-CoV-2 infection in vivo will require further study. It is also important to consider that reported COVID-19 disease manifestations are diverse and may be explained by factors other than vascular leak, including pneumocyte damage resulting from immune cell infiltration and viral infection. Understanding the relative contribution of vascular leak to COVID-19 disease severity will undoubtedly be complicated but is nevertheless a critical question.

It is important to note that our working hypothesis is not that S mediates disease pathogenesis alone, but rather that the reversible vascular leak triggered by S may serve to promote viral dissemination of

SARS-CoV-2 into distal tissues of infected patients, which could lead to severe disease manifestations. However, although we observe significant vascular leak in mice administered S alone, they do not overtly display signs of morbidity. Importantly, our findings suggest that the amounts of S circulating in patients following COVID vaccination (pg/mL levels) are too low to trigger vascular leak given that our phenotype requires ng-μg/mL levels that mimic the levels observed during severe COVID-19 cases[56,60]. Taken together, our study and available literature[76] indicate that S-mediated vascular leak would not result from COVID-19 vaccination and therefore is not correlated with any vaccine adverse events.

In sum, our study reveals the role of S in COVID-19-associated vascular leak and provides mechanistic insight into how S mediates this process independently from viral infection and the ACE2 receptor. Although much work remains to be done to fully understand the structural basis of this mechanism as well as the implications for this pathway in SARS-CoV-2 infection and disease in humans, this work provides a foundation for future investigations by beginning to define the contribution of S to COVID-19-associated vascular leak.

## Methods
### Mice
Six- to eight-week-old wild-type C57BL/6 J and K18-hACE2 [B6.Cg-Tg(K18-ACE2)2Prlmn/J] mice of both genders were purchased from the Jackson Laboratory and housed at the University of California, Berkeley Animal Facility under specific pathogen-free conditions. Mice were housed in temperature-controlled environments on a 12-hour light and dark cycle, with food and water provided ad libitum. All experiments and procedures were pre-approved by the UC Berkeley Animal Care and Use Committee, Protocol AUP-2014-08-6638-2 and AUP-2020-07-13458 and conducted in compliance with Federal and University regulations.

### Cell lines
HEK293T cells used for lentivirus production were obtained from ATCC and maintained in DMEM, high glucose, and GlutaMAX™ (Gibco) supplemented with 10% FBS (Corning) and 1% penicillin/streptomycin (Gibco) (D10 medium) at 37 °C with 5% $CO_2$. Calu-3 human lung epithelial cells were obtained from the UC Berkeley Cell Culture Facility and maintained in D10 medium at 37 °C with 5% $CO_2$. Human pulmonary microvascular endothelial cells (HPMEC) [line HpMEC-ST1.6 R] were a gift from Dr. J.C. Kirkpatrick at Johannes Gutenberg University, Germany. These cells were isolated from an adult human male donor, and an immortalized clone was selected that displayed all major phenotypic makers of pulmonary endothelial cells[77]. HPMECs were cultured in endothelial cell growth basal medium 2 supplemented with an Endothelial Cell Growth Medium-2 (EGM-2™) supplemental bullet kit (Lonza). Cells were maintained at 37 °C with 5% $CO_2$. This study produced several genetically modified versions of these parental cells, including HPMEC overexpressing the human ACE2 (hACE2) gene (HPMEC/ACE2) as well several knockout cell lines. The

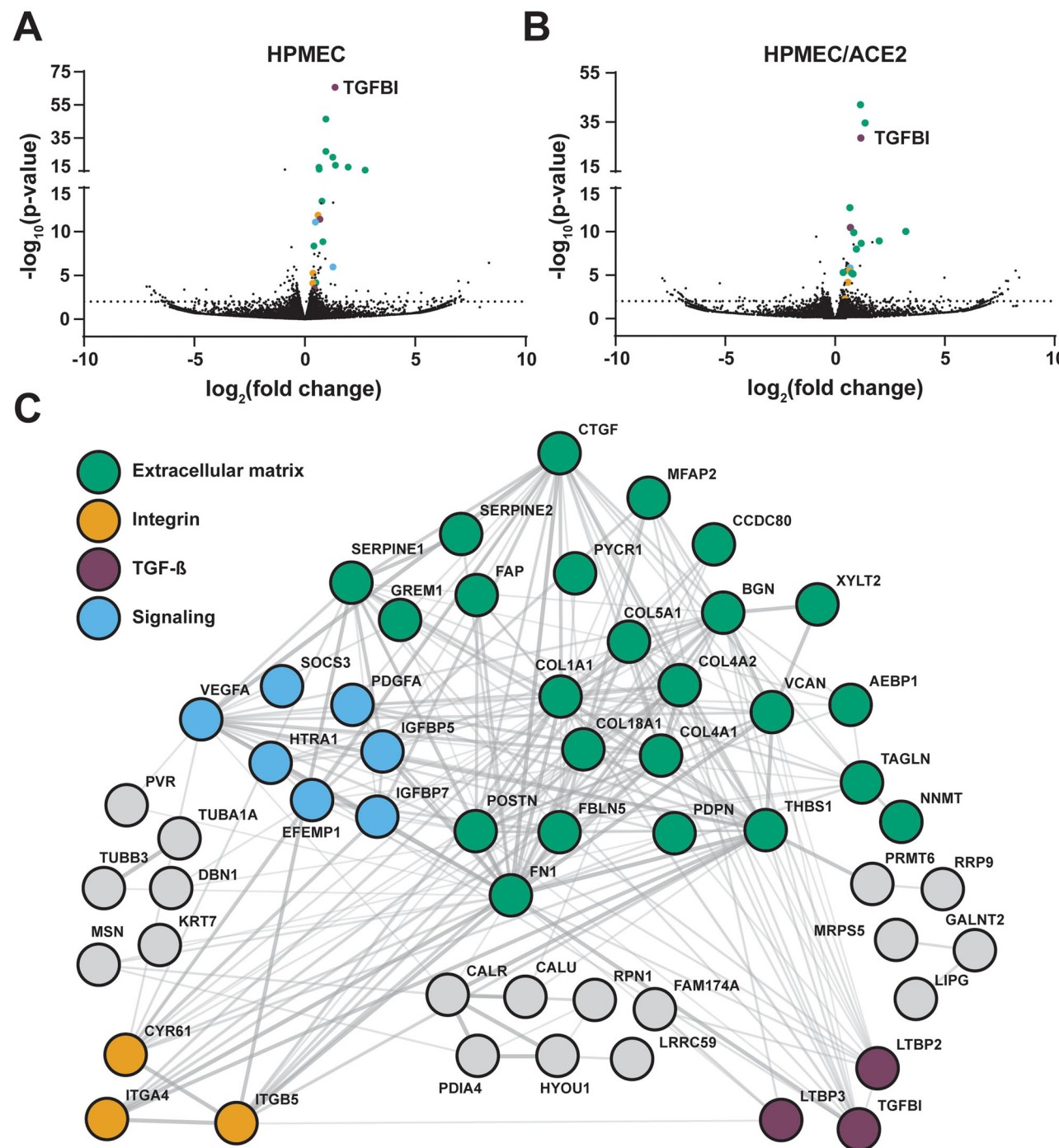

**Fig. 5 | RNA-Sequencing analysis of SARS-CoV-2 S-treated HPMEC and HPMEC/ACE2. A** A volcano plot of differentially expressed genes (DEGs) detected in HPMECs treated with 10 µg/mL of SARS-CoV-2 S at 24 hpt. **B** Same as **A** but displaying DEGs from HPMEC/ACE2 treated with 10 µg/mL SARS-CoV-2 S. Dotted lines indicate the threshold for significance. Statistical significance of DEGs was determined using a Wald test and a Benjamini-Hochberg (BH) *p*-value adjustment. **C** A STRING protein-protein interaction network of DEGs identified between S-treated and untreated conditions for HPMEC.

hACE2 encoding plasmid was a gift from Hyeryun Choe (Addgene plasmid #1786; http://n2t.net/addgene:1786; RRID:Addgene_1786)[70]. Vero-E6 cells were used for SARS-CoV-2 titration and maintained in D10 media at 37 °C with 5% $CO_2$.

### Recombinant proteins
Sequences encoding for full length, stabilized SARS-CoV-2 S ectodomain and RBD (based on the Wuhan-Hu-1 sequence)[78] were expressed and purified from stably transformed 293 cells, as previously reported[79]. Purified proteins were formulated at -1 mg/mL in PBS and

stored in aliquots at −80 °C. Recombinant DENV serotype 2 NS1 was purchased from the Native Antigen Company and characterized previously (Dengue virus serotype 2 NS1 [accession # P29990.1, Thailand/ 16681/84])[36]. SARS-CoV-2 S stabilized trimers from diverse viral variants were purchased from the Native Antigen Company including B.1.1.7 (Alpha variant, product #REC31924), B.1.351 (Beta variant, product #REC31963), B.1.1.28/P.1 (Gamma variant, product #REC31944), B.1.617.2+AY.1+AY.2+AY.3 (Delta variant, product #REC31975), and B.1.1.529 (Omicron variant, product #REC32008). Recombinant VEGF (V7259) and TGF-β1 (100-21) was purchased from Sigma and

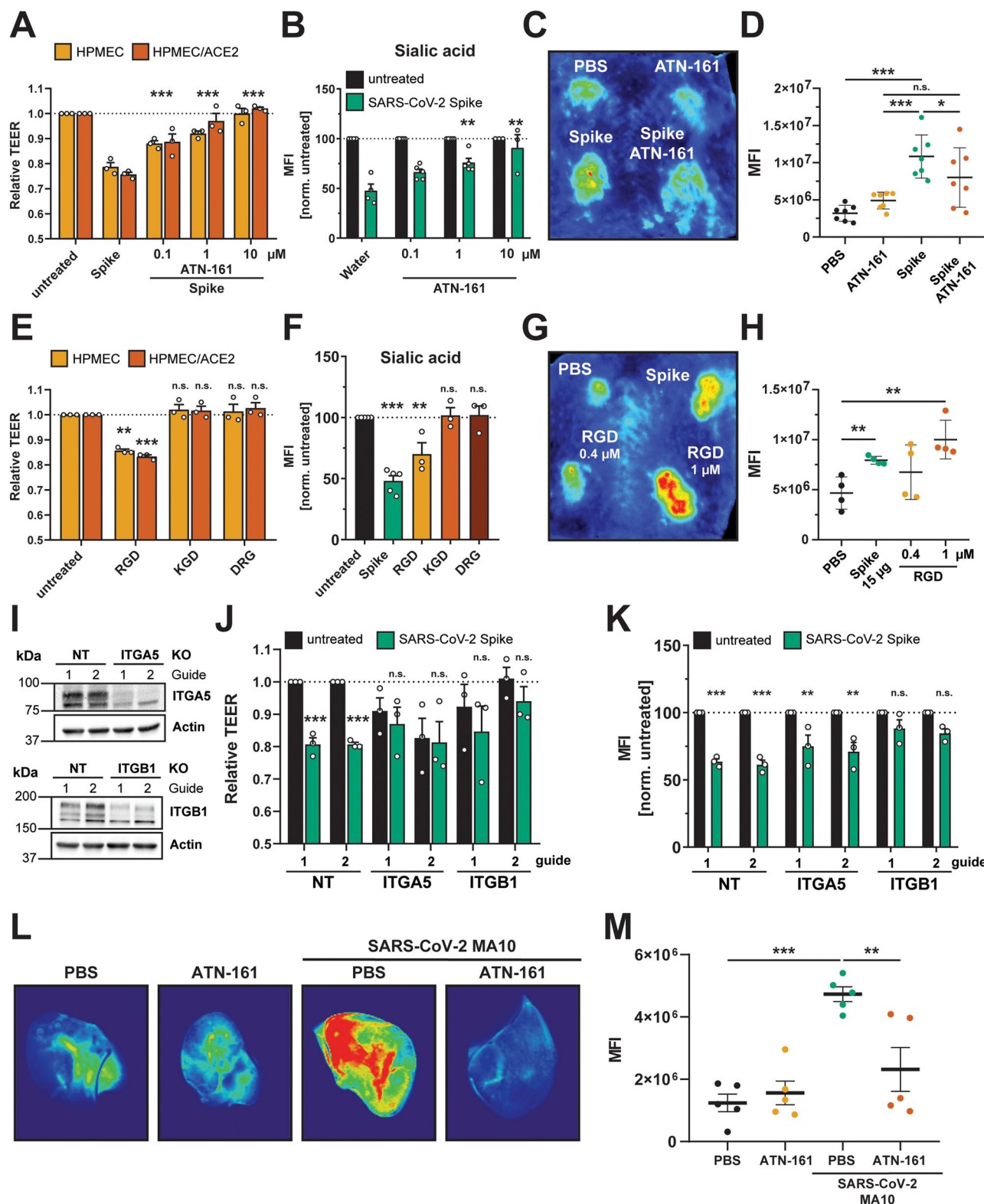

PeproTech, respectively, and resuspended/stored in accordance with the manufacturer's instructions.

## Vesicular stomatitis virus (VSV) spike pseudotype virus production and infection

SARS-CoV-2 S pseudotyped virions were produced using the VSV-ΔG-rLuc system, originating from the VSV Indiana serotype full-length complementary DNA clone in which the G glycoprotein was exchanged for a *Renilla* Luciferase reporter gene. In brief, 15-cm² dishes of HEK293T cells were transfected with a plasmid encoding SARS-CoV-2 S (Wuhan-Hu-1, Accession #QHD43416.1) using 45 μg total DNA. At 24 h post-transfection, VSV-G expressing VSV-ΔG-rLuc pseudotyped virions were used to infect the transfected HEK293T cells. Medium was harvested 48 hours post-transfection and ultracentrifuged for 1.5 h at 110,000 × *g* in a 20% sucrose cushion in NTE buffer (150 mM NaCl, 40 mM Tris-HCl, and 1 mM EDTA, pH 8.0). For viral resuspension, NTE

**Fig. 6 | Integrins are required for SARS-CoV-2 S-mediated and SARS-CoV-2 infection-mediated barrier dysfunction. A** TEER inhibition assay of HPMECs and HPMEC/ACE2 monolayers treated with S (10 μg/mL) and the indicated concentrations of the integrin inhibitor ATN-161. TEER readings were taken 24 hpt with ATN-161 and S added simultaneously to cells. Data are from n = 3 biological replicates. **B** EGL inhibition assay detecting sialic acid on the surface of HPMEC monolayers treated with S and ATN-161 as in **A**. Data are from at n = 4 (water), n = 5 (ATN-161 0.1 μM and 1 μM) and n = 3 (ATN-161 10 μM) biological replicates. **C** Representative back from an intradermal leak assay of mice with the indicated treatments; S (15 μg) and ATN-161 (1 μM) injected simultaneously. **D** Quantification of **C** from n = 7 mice. **E** TEER assay of HPMEC and HPMEC/ACE2 monolayers treated with the indicated peptides at 0.4 μM. TEER readings were taken 24 hpt. Data are from n = 3 biological replicates. **F** Same as **E**, but an EGL assay detecting sialic acid on the surface of HPMEC monolayers. Data are from n = 3 biological replicates. **G** Representative back from an intradermal leak assay of mice with the indicated treatments with S (15 μg) and the indicated doses of RGD peptide. **H** Quantification of **G** from n = 4 mice. **I** Western blot analysis of HPMEC transduced with the indicated lentivirus-encoding guide RNA. Actin was used as a loading control. Data are one

representative experiment from n = 3 biological replicates. **J** TEER assay of HPMEC transduced with lentivirus-encoding guide RNAs targeting the indicated genes as in **I**. Cells were treated with 10 μg/mL of S, and TEER was read 24 hpt. Data are from n = 3 biological replicates. **K** EGL assay detecting sialic acid on the cell surface of transduced HPMEC as in **J**. Data are from n = 3 biological replicates. **L** Representative lung images from C57BL/6 mice infected with $2 \times 10^4$ PFU of SARS-CoV-2 mouse-adapted strain (MA-10) for 7 days. Mice were administered 10 mg/kg ATN-161, or a vehicle control, intraperitoneally daily (8 doses total). Mice were administered a dextran-680 tracer intravenously on day 7 post-infection. Lungs were collected 2 hours after dextran-680 administration and fixed overnight in formalin, and the fluorescence accumulation was measured with a fluorescent scanner. **M** Quantification of **L** from n = 5 mice. MFI is mean fluorescence intensity. Dotted lines are the normalized untreated control conditions. All data are as plotted as mean +/− SEM, with *$p < 0.05$, **$p < 0.01$, ***$p < 0.001$, and n.s. $p > 0.05$ by One-Way ANOVA with Tukey's Multiple comparisons test except for (**D**, **H**, and **J**) which were analyzed by two-sided unpaired t-test. Statistics in panels **E**, **F**, **J**, and **K** are comparisons to untreated controls and in panels **A** and **B** are comparisons to the S-only control condition. Source data are provided as a Source Data file.

Buffer was supplemented with 5% sucrose, and viral aliquots were stored at −80 °C.

## SARS-CoV-2 stock production and infection
Stocks of SARS-CoV-2 were produced as previously described[80]. In brief, the USA-WA1/2020 or mouse-adapted MA10 strain of SARS-CoV-2 were obtained from BEI Resources or Dr. Ralph S. Baric at the University of North Carolina, Chapel Hill[81] respectively, and passed through a 0.45 μM syringe filter. Five μL of these filtered stocks were added to T-175 flasks of Vero-E6 cells to produce virus passage 1. Cytopathic effect (CPE) was monitored daily and flasks were frozen down when ~70% cytopathic effect was evident (~48 hpi). Thawed lysates were then collected, and cell debris was pelleted at 3000 rpm for 20 min. Clarified viral supernatant was then aliquoted, and infectious virus was quantified by $TCID_{50}$. To produce passage 2, SARS-CoV-2 working stocks, 5 μL of the passage 1 stock was inoculated onto T175 flasks of Vero-E6 cells as described above. Viral titers obtained ranged from $1 \times 10^6 – 5 \times 10^6$ $TCID_{50}$ units/mL. For mouse infection, 8- to 10-week-old K18-hACE2 or C57BL/6 male mice were anesthetized using isoflurane and intranasally inoculated with the strain and dose of SARS-CoV-2 indicated in the figures.

## Lentivirus production and transduction
Lentivirus particles encoding human ACE2 or CRISPR guide RNAs targeting the genes indicated in the figures were produced using a second-generation lentivirus system as reported previously[82]. In brief, lentivirus vectors were transfected into 293 T cells using a lipofectamine 3000 transfection protocol according to the manufacturer's instructions, along with a packaging vector (psPAX2) and a pseudotyping vector (pMD2.G). Medium was replaced on transfected cells 12 hours post-transfection. Lentivirus released into the medium of transfected cells was collected at 24-, 36-, and 48-hours post-transfection, and pooled lentivirus-containing media was filtered through a 0.45 μM syringe filter (Millipore). Target HPMECs were incubated with lentivirus for 48 hours and then selected with 2 μg/mL puromycin (Sigma) for 3 passages.

## Trans-epithelial/endothelial electrical resistance assay
Epithelial and endothelial barrier disruption (hyperpermeability) was measured through a Trans-Epithelial/Endothelial Electrical Resistance (TEER) Assay as previously described[36]. In brief, $6 \times 10^4$ HPMEC or $2 \times 10^5$ Calu-3 were seeded in 300 μL into the apical chambers of 24-well transwell polycarbonate membrane inserts (Transwell permeable support, 0.4 μM, 6.5 mm insert; Corning) and 1.5 mL of medium was added to the basolateral chamber. Medium was changed daily from

both the apical and basolateral chambers until cells formed a complete monolayer measured through maximal barrier resistance (~3 days for HPMEC and ~15 days for Calu-3). On the day of the experiment, TEER of each transwell was measured to ensure that cell resistance levels were at a minimum value and roughly equivalent (within 5 ohms). Transwells with outlier resistance values were excluded from the experiment. Treatments (indicated in the figures) were added to the apical chamber of the transwells. Electrical resistance values were measured in ohms (Ω) at the time-points indicated in the figures using an Epithelial Volt Ohm Meter (EVOM) with a "chopstick" electrode (World Precision Instruments). Inserts with untreated cells, as well as inserts with no cells containing medium alone (blank), were used as negative controls to calculate the baseline electrical resistance. Relative TEER was calculated as a ratio of resistance values as (Ω experimental condition-Ω blank)/(Ω untreated cells-Ω blank).

## Endothelial/epithelial glycocalyx layer (EGL) disruption assays
To measure the capacity of protein treatments to mediate EGL disruption, $6 \times 10^4$ HPMEC or $2 \times 10^5$ Calu-3 cells were seeded on 0.2% gelatin- (Sigma) coated glass coverslips in 24-well plates. Cells were allowed to form a fully confluent monolayer over three days (HPMEC) or ~15 days (Calu-3), with medium changes every other day. On the day of the experiment, the treatments indicated in the figures were added directly to wells. Treatments and cells were allowed to incubate for the indicated times (generally 24 hours for S treatments) and then cells were washed twice with 1x PBS and fixed with 4% formaldehyde/PBS (Thermo Fisher Scientific). Coverslips were mounted onto microscope slides on a drop of ProLong Gold (Thermo Fisher Scientific) and imaged using a Zeiss LSM 710 inverted confocal microscope (CRL Molecular Imaging Center, UC Berkeley). EGL disruption was assessed by monitoring surface levels of sialic acid on the cell surface using the sialic acid-specific lectin, Wheat Germ Agglutinin (WGA) conjugated to Alexa Fluor 647 (Thermo Fisher Scientific, W32466), hyaluronic acid (Abcam ab53842), heparan sulfate (amsbio, clone F58-10E6, 370255-s), or chondroitin sulfate (Thermo Fisher Scientific, clone CS-56, ma1-83055). EGL disrupting enzyme expression were assessed on saponin permeabilized cells for hyaluronidase (Abcam, clone PH20, ab196596) and neuraminidase 2 (Thermo Fisher Scientific, pa5-35114). For staining, cells were live cell stained at 100 μg/mL of WGA-647 diluted in the medium of live cells 1 hour pre-fixation. All other GAGs (HA, HS, CS) were stained on fixed cells with the coverslip turned onto a 15-μl drop of antibody containing staining buffer. Nuclei were stained by adding Hoechst 33342 (Immunochemistry) at a dilution of 1:200. All microscopy images were captured at 20x magnification.

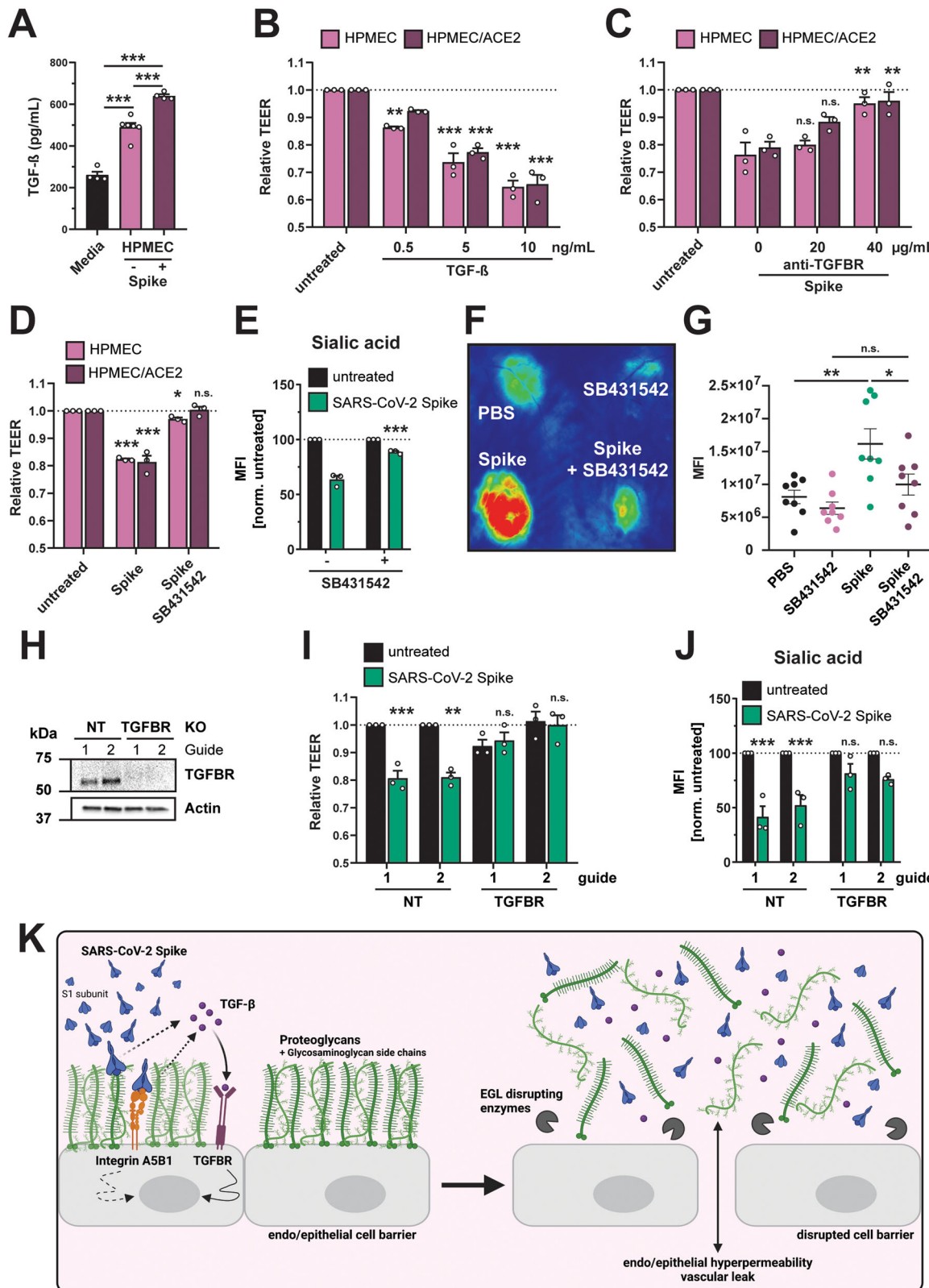

## In vivo dermal vascular leak assay

A murine dermal leak model was utilized to investigate vascular leak triggered by the S glycoprotein as previously described[36]. In brief, the dorsal dermises of 6-7-week-old WT C57BL/6 J female mice (purchased from Jackson Labs) were shaved. After 3-4 days, the indicated treatments (typically 10 μg of SARS-CoV-2 S) were injected intradermally (ID) into discrete spots in the shaved mouse dermis (50 μL/

injection site). Immediately following ID injections, 150 μL (25 μg total) of 10-kDa dextran conjugated to Alexa Fluor 680 (Sigma) was delivered intravenously (IV). Two hours post-injections, mice were euthanized, and the dorsal dermises were removed and placed in petri dishes. Fluorescence signal accumulation in the mouse dermises was visualized using a fluorescence scanner (LI-COR Odyssey CLx Imaging system) at a wavelength of 700 nm. Leakage was

**Fig. 7 | SARS-CoV-2 S triggers production of TGF-β, and TGF-β signaling is required for S-mediated barrier dysfunction. A** Commercial ELISA detecting TGF-β in medium without cell conditioning (Media), medium from untreated HPMEC, and medium from HPMEC treated with 10 μg/mL SARS-CoV-2 S. Data are from $n = 4$ (media and HPMEC + S) and $n = 6$ (HPMEC) biological replicates. **B** TEER assay measuring the effect of recombinant TGF-β on barrier function of HPMEC at the indicated concentrations. TEER readings were taken 24 hpt. Data are from $n = 3$ biological replicates. **C** TEER assay measuring the capacity of an anti-TGFBR antibody, at the indicated concentrations, to abrogate S-mediated endothelial hyperpermeability (S at 10 μg/mL) of HPMECs and HPMEC/ACE2. TEER readings were taken 24 hpt. Data are from $n = 3$ biological replicates. **D** TEER assay measuring the capacity of TGFBR inhibitor SB431542 (1 μM) to inhibit S (10 μg/mL) function. Data are from $n = 3$ biological replicates. **E** Same as **D**, except an EGL assay measuring sialic acid. Data are from $n = 3$ biological replicates. **F** Representative back from an intradermal leak assay of mice with the indicated treatments; S (15 μg) and SB431542 (1 μM) were injected simultaneously. **G** Quantification of **F** from $n = 8$ mice. **H** Western blot analysis of HPMECs transduced with lentivirus-encoding

guide RNAs targeting the indicated genes. Actin was used as a loading control. Data are one representative experiment from $n = 3$ biological replicates. **I** TEER assay on the same HPMEC as in **H** treated with 10 μg/mL of S and measured 24 hpt. Data are from $n = 3$ biological replicates. **J** EGL disruption assay on HPMECs from **H**, treated with 10 μg/mL S and imaged 24 hpt. Control guide data from this panel are from the same experiment as Fig. 4G. Data are from $n = 3$ biological replicates. **K** Graphical abstract summarizing the ACE2-independent pathway by which SARS-CoV-2 S triggers barrier dysfunction. Solid lines represent steps with direct experimental evidence, while dotted lines represent hypothesized steps. For all figures, dotted lines in graphs are the normalized untreated control conditions. MFI is mean fluorescence intensity. All data are plotted as mean + /− SEM with *$p < 0.05$, **$p < 0.01$, ***$p < 0.001$, and n.s. $p > 0.05$ by One-Way ANOVA with Tukey's Multiple comparisons test except for (**G**) which was analyzed by two-sided unpaired $t$-test. Statistics in panels **B**, **D**, **I**, and **J** are comparisons to untreated controls and in panels **C** and **E** are comparisons to the S-only control condition. Source data are provided as a Source Data file.

quantified around the injection sites using Image Studio software (LI-COR Biosciences). For small molecule inhibitor experiments, the inhibitor was mixed with SARS-CoV-2 S for 30 minutes before the ID injection.

### In vivo systemic leak
To investigate systemic vascular leak triggered by SARS-CoV-2 S or SARS-CoV-2 infection, we conducted a modified systemic vascular leak assay as described previously[32]. In brief, 50 μg of SARS-CoV-2 S, OVA or SARS-CoV-2 viral stocks (indicated in the figures) were administered to 6-7-week-old WT C57BL/6 J female mice (purchased from Jackson Labs) intranasally. Twenty-two hpt and seven days post infection as indicated, mice received an IV injection of 10-kDa dextran conjugated to Alexa Fluor 680 (150 μL, 170 μg/mL; Sigma). This tracer dye was allowed to circulate in mice for 2 hours, at which time mice were euthanized and organs (lungs, spleen, liver, small intestine, and brain) were collected and placed on petri dishes. For SARS-CoV-2 infections organs were fixed in 10 mL of 10% neutral buffered formalin solution (Sigma) overnight prior to imaging. Fluorescence signal accumulation in organs was visualized using a fluorescence scanner (LI-COR Odyssey CLx Imaging system) at a wavelength of 700 nm. Leakage was quantified using Image Studio Lite software version 5.2 (LI-COR Biosciences).

### Hematoxylin and eosin (H&E) staining
Histology and H&E staining was performed by HistoWiz Inc. (histowiz.com) following a standard protocol and automated workflow. Organs were collected and fixed in 10% neutral buffered formalin solution overnight at UC Berkeley and then shipped to HistoWiz where they were processed and embedded in paraffin and 4 μm sections were prepared. After H&E staining, sections were dehydrated and film coverslipped using a TissueTek-Prisma and Coverslipper (Sakura). Whole slide scanning (40×) was performed on an Aperio AT2 microscope (Leica Biosystems).

### EGL enzyme preparation and digestion
To test for a contribution of glycan components to S-mediated endothelial dysfunction, recombinant enzymes were used to digest specific glycan components, including hyaluronic acid, heparan sulfate, sialic acid, and chondroitin sulfate. Recombinant heparin lyases I and III were obtained from IBEX and HPMECs were treated with 5 mU/mL of each, recombinant hyaluronidase (Sigma, H3506) was treated at 10 μg/mL, recombinant neuraminidase (Sigma, N2876) was treated at 1 U/mL, and recombinant chondroitinase ABC (Sigma, C3667) was treated at 25 mU/mL. All enzymes were added to HPMECs simultaneously with S and TEER/EGL assays were conducted 24 hpt as described above.

### CRISPR-Cas9 knockout
To produce gene-specific knockout cell lines, we utilized a CRISPR-Cas9 pipeline based on the lentiCRISPR v2 lentivirus construct obtained from Feng Zhang (Addgene plasmid # 52961; http://n2t.net/addgene:52961; RRID:Addgene_52961), as previously described[83]. In brief, guide RNA targeting sequences were selected from the Brunello CRISPR KO guide library[84] and cloned into the lentiCRISPR v2 plasmid. Guide RNA sequences utilized in this study are summarized in Table S3. Lentivirus was produced as described above, and HPMEC were transduced and selected in EGM-2 medium containing 2 μg/mL of puromycin and passaged three times in selection. Polyclonal populations of cells were characterized for functional knockout through either a Western blot to measure protein expression or through HS staining by IFA to confirm function (for HS biosynthetic pathway genes).

### RNA sequencing
To characterize the transcriptional response of cells treated with SARS-CoV-2 S, we conducted RNA-Sequencing. In brief, HPMECs and HPMEC/ACE2 were treated with 10 μg/mL SARS-CoV-2 S, and cell lysates were collected in TRI reagent (Sigma) 24 hours post-treatment. Total RNA was extracted using a Direct-zol RNA miniprep kit (Zymo Research) per the manufacturer's instructions. RNA was quantified using a Qubit Flex Fluorometer (ThermoFisher), and quality was measured by Bioanalyzer (RNA Pico; Agilent). 1 μg of RNA was used for library preparation with the SMARTer Stranded Total RNA Sample Prep Kit - HI Mammalian (Takara Bio), following the manufacturer's instructions. Quality of prepared libraries was evaluated by Bioanalyzer (High Sensitivity DNA; Agilent) and sequenced on the NovaSeq 6000 (Illumina) using S4 flow cell and 150-base pair paired-end sequencing at the UCSF Center for Advanced Technology. Following sequencing of sample libraries, quality control was performed on the fastq files to ensure that sequencing reads met preestablished cutoffs for number of reads and quality using FastQC (version 0.11.8)[85] and MultiQC (version 1.8)[86]. Quality filtering and adapter trimming were performed using BBduk tools (version 38.76, https://sourceforge.net/projects/bbmap). Remaining reads were aligned to the ENSEMBL GRCh38 human reference genome assembly (release 33) using STAR (version 2.7.0 f)[87], and gene frequencies were counted using featureCounts (version 2.0.0) within the Subread package[88]. Comparative analysis of DEGs was performed using a negative binomial distribution model used by DESeq2 (version 1.28.1)[89] as implemented in R (version 4.0.3). All genes passing a Benjamini-Hochberg (BH)-adjusted P value threshold of 0.05 were included. Hierarchical clustering of DEGs and visualization were performed using the ComplexHeatmap (version 2.4.2) and pheatmap package (version 1.0.12). The clustering method used is complete linkage, and clusters are based on Euclidean distance. Identified DEGs were analyzed by STRING to identify predicted

protein-protein interaction networks as well as enriched pathway analysis (https://string-db.org/).

## TGF-β enzyme linked immunosorbent assay (ELISA)

To measure levels of TGF-β1 in the supernatants of S-treated cells, we used the Human TGF-beta 1 DuoSet ELISA (DY240, R&D Systems). In brief, cell supernatants were collected 24 h after treatment with SARS-CoV-2 S and activated with 1 N HCl in order to detect immunoreactive TGF-β1. After pH neutralization, samples and recombinant TGF-β1 standards were transferred to ELISA plates coated with mouse anti-human TGF-β1 capture antibody and incubated for 2 h at room temperature. Afterwards, plates were incubated with biotinylated chicken anti-human TGF-β1 detection antibody and then with streptavidin-horseradish peroxidase (HRP) for signal detection with tetra-methylbenzidine (TMB) substrate. The optical density of each well was determined using a microplate reader set to 450 nm. TGF-β1 levels were determined by interpolation from four-parameter logistic (4-PL) standard curves.

## Size-exclusion chromatography with multi-angle light scattering (SEC-MALS)

To characterize the purity and oligomeric state of soluble coronavirus S and RBD, purified proteins were injected onto a SRT SEC-1000 column (4.6 × 300 mm, Sepax) at 0.35 mL/min in PBS (for S) or a Superdex 200 Increase column (3.2 × 300 mm, Cytiva) at 0.15 mL/min in PBS (for RBD). The column and the entire SEC-MALS system, which includes a 1260 Infinity II HPLC (Agilent), a miniDAWN TREOS II MALS detector (Wyatt) and a Optilab T-rEX refractive index detector (Wyatt), were equilibrated in PBS for at least 24 h prior to analysis. Molecular weights of S and RBD were determined using Astra (Wyatt).

## Small molecule inhibitors, peptides, and antibodies

The following small molecule inhibitors and antibodies were used for TEER and EGL inhibition assays: ATN-161 (Sigma, SML2079), SB 431542 hydrate (Sigma, S4317), heparin (Sigma, H3393), anti-Spike (Genetex, 1A9, GTX632604), anti-Spike (Absolute Antibody, CR3022), rabbit anti-TGFBR1 (Thermo Fisher Scientific, PA5-32631). All chemicals and antibodies were resuspended and utilized per the manufacturer's instructions. The RGD, KGD, and DRG peptides were synthesized by Sigma.

## SDS-PAGE and western blot

Recombinant proteins or cell lysates were collected in protein sample buffer (0.1 M Tris pH 6.8, 4% SDS, 4 mM EDTA, 286 mM 2-mercaptoethanol, 3.2 M glycerol, 0.05% bromophenol blue) and resolved by SDS-PAGE. Proteins were then transferred to nitrocellulose membranes and probed with primary antibodies diluted in PBS with 0.1% Tween20 (PBST) containing 5% skim milk. Membranes and antibodies were incubated overnight rocking at 4 °C. The next day, membranes were washed three times with PBST before being probed with HRP-conjugated secondary antibodies diluted at 1:5,000 in 5% milk in PBST at room temperature for 1 hour. Membranes were then washed with PBST three more times before being developed with homemade ECL reagents and imaged on a ChemiDoc system with Image Lab software version 6.01 (Bio-Rad). The following antibodies were used in this study: goat anti-ACE2 (R&D Systems, AF933), rabbit anti-integrin alpha 5 (Abcam, ab150361), rabbit anti-ITGB1 (Thermo Fisher Scientific, PA5-29606), rabbit anti-Heparanase 1 (Abcam, EPR22365-230, ab254254), rabbit anti-MMP-9 (Cell Signaling Technology, #3852), mouse anti-TACE/ADAM17 (Santa Cruz Biotechnologies, B-6, sc-390859), mouse anti-Cathepsin L (Thermo Fisher Scientific, 33-2, BMS1032), rabbit anti-TGFBR1 (Thermo Fisher Scientific, PA5-32631), mouse anti-His (MA1-21315, Thermo Scientific), mouse anti-β-actin HRP (Santa Cruz Biotechnologies, sc-47778 HRP), goat anti-mouse HRP (Biolegend,

405306), donkey anti-rabbit HRP (Biolegend, 406401), donkey anti-human HRP, Biolegend, 410902). All primary antibodies were used at a dilution of 1:1000 and all secondary antibodies were used at a concentration of 1:5000.

## Statistics

All data were plotted and quantitative analyses performed using GraphPad Prism 8 software. Experiments were repeated at least 3 times, except when indicated otherwise. Experiments were designed and performed with both positive and negative controls, which were used for inclusion/exclusion determination. Researchers were not blinded during experiments. For immunofluorescence microscopy experiments, images of random fields were captured. For all experiments with quantitative analyses, data are displayed as mean ± SEM. Statistical tests used in this study include ANOVA analysis with multiple comparisons test as well as $t$-tests, as indicated in the figures. Resultant $p$-values from the above statistical tests are displayed as n.s., not significant $p > 0.05$; *$p < 0.05$; **$p < 0.01$; ***$p < 0.001$. All statistics not indicated are not significant.

## Reporting summary

Further information on research design is available in the Nature Portfolio Reporting Summary linked to this article.

## Data availability

All raw data associated with the figures are either included in this submission and/or available upon request. RNA-Seq raw data are uploaded to NCBI SRA as part of BioProject accession # PRJNA807823 (https://www.ncbi.nlm.nih.gov/bioproject/?term=PRJNA807823). Source data are provided with this paper.

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

## Acknowledgements

We thank Dr. Michael S. Diamond (Washington University School of Medicine) and Dr. Peter Laing (Excivion Ltd) for helpful discussion. Confocal imaging experiments were conducted on a Zeiss LSM 710 at the CRL Molecular Imaging Center at UC Berkeley, which is supported by the Gordon and Betty Moore Foundation. BioRender was used for creation of Fig. 1A and 7K. This work was supported by NIAID/NIH grants R01 AI24493 (E.H.), R21 AI146464 Supplement (E.H.), and a Fast Grant from Emergent Ventures (E.H.). J.D.E. was supported by NSF grant RAPID 201989 and NIH/NHLBI grant HL131474. H.A.C. was supported by R01 AI109022 and a Fast Grant (Emergent Ventures). A.M.N. was supported by the IGI, Fast Grants (Emergent Ventures), and an anonymous donor to UC Berkeley. S.B.B. was supported in part as an Open Philanthropy Awardee of the Life Sciences Research Foundation.

## Author contributions

S.B.B., F.T.G.S., P.R.B., and E.H. conceived the study. J.E.P., S.A.S., and E.H acquired funding. S.B.B., F.T.G.S., L.V.T., F.P., C.Z., R.R., S.F.B., T.S.P., D.R.G., B.C.R., V.S., Ca.M.W., N.T.N.L., M.P.W., Co.M.W., D.R.S., T.M.C., Y.A.S., D.M.F., and V.O. conducted experiments. S.B.B., F.T.G.S., L.V.T., F.P., C.Z., R.R., S.F.B., T.S.P., D.R.G., D.R.S., T.M.C., V.O., P.R.B., and E.H. designed experiments. S.B.B., A.M.N., S.A.S., H.C.A., J.D.E., C.Y.C., J.E.P., and E.H. provided supervision. R.S.B. provided critical resources. S.B.B., F.T.G.S., L.T., F.P., D.R.G., V.S., and E.H. designed figures. S.B.B., F.T.G.S., and E.H. wrote the original draft of this manuscript and S.B.B., F.T.G.S., F.P., P.R.B., and E.H edited subsequent drafts.

## Competing interests

The authors declare no competing interests.
