## [Peer Review File · Nature Communications]

Reviewer comments, first round –

Reviewer #1 (Remarks to the Author):

The manuscript "SARS-CoV-2 triggers barrier dysfunction and vascular leak via integrins and TGF-B signaling" by Eva Harris and colleagues is an exceptional manuscript that investigated the mechanisms underlying SARS-CoV-2 Spike protein-induced hyperpermeability.

The manuscript is well written and provides both *in vitro* and *in vivo* assays. Authors have initially investigated different types of viral proteins and their effect on endothelial monolayers and later identified critical necessary pathways for Spike protein-induced endothelial dysfunction. The manuscript later discusses the relevance of the cross-talk between endothelial cells and ECM proteins in regards to SP pathogenicity and identify TGF-B as a critical signaler of injury.

The data is presented in a logic progression and the number of experiments is adequate. The reading is thus pleasant and of great interest due to the ongoing Pandemic.

Few minor comments:

- The first evidence of Spike protein-induced endothelial barrier dysfunction was provided by Prof. Ramirez et al. (<https://pubmed.ncbi.nlm.nih.gov/33053430/>). The manuscript would be improved by adding this important information.

- In that manuscript and in a more recent one (<https://www.frontiersin.org/articles/10.3389/fphys.2022.812199/full>), SP was able to elicit barrier dysfunction with a the dose of 10-20 nM, while in this paper a much higher amount was used (ug/ml). Authors, should explain their decision to use such a high concentration of spike protein and discuss it within the text.

Reviewer #2 (Remarks to the Author):

The pathophysiology of clinical COVID-19 disease suffice to say is complicated. Many reports have indicated impacts of the innate immune response, inflammatory responses and the over-production and contributions of cytokines, including IL-6. In this report Biering et al demonstrate that components of the SARS-CoV-2 Spike protein can bind to GAGs and integrins on the cell surface and induce a TGF-b response resulting in barrier dysfunction and vascular leakage. This work builds upon the previous excellent studies done by this lab on the NS1 glycoprotein of dengue virus (DENV), a well known hemorrhagic virus. The approach and methodologies have been well established by this lab in demonstrating vascular leakage by the DENV NS1 and equally here they are performed to a high standard and the results are of a very conclusive and high standard. Albeit the focus is primarily on the Spike protein alone and not on virus infection itself. The results are of significance particularly with the administration of whole Spike and RBD components as vaccine antigens. Certainly the methodology is sound and the conclusions drawn from these results are appropriate. The biggest difficulty is the correlation between expression of the Spike proteins and the actual events happening during infection both *in vitro* and *in vivo*.

Major issues to amend:

1. The results need to be compared to infection. This can be done in both the HPMEC and HPMEC/ACE2 cells. One would expect that infection of HPMEC should induce the same response as protein alone in the absence of active infection and replication. This is also true in the Calu-3 cells where UV-inactivated virus should be used.
2. Some consideration should be made when comparing the relative dose of Spike vs NS1 vs VEGF. All different doses were used, so firstly why and is this related to molarity? What is the basis for using some much protein (ie. 10ug/mL) when it was suggested that only 250 ng/mL is required (line 311). Was 250ng/mL tried and what was the outcome?
3. Infection of mice should also be evaluated by histology for barrier dysfunction, this can be achieved with Evans Blue or similar

4. The authors show that the RGD domain of Spike is sufficient to induce leakage, so then is this applicable to all other viral structural proteins that contain this motif? Importantly Adenovirus 5 that is used as a vaccine to deliver Spike. This is essence should induce dysfunction via both the virus vector and the encoded Spike protein.

5. A report published by Rauti et al (2021) eLife expressed all of the SARS-CoV-2 proteins in HUVECs and performed similar assays to those described here. However, Spike was not observed to induce leakage in their studies, many of the other non-structural proteins did. How can such differences be explained? Did the authors also try HUVECs? Although the point below may counter this

6. Would also comment on the recent publication by Robles et al (2022) JBC who have shown that Spike, RBD and RGD induce hyperpermeability in HUVECs via $\alpha 5 \beta 1$ integrin that can be restricted by treatment with Volciximab and ATN-161 so some novelty is lost in this report

Reviewer #3 (Remarks to the Author):

In "SARS-CoV-2 Spike triggers barrier dysfunction and vascular leak via integrins and TGF- β signaling," Biering et al. describe a study in which a novel role for the Spike (S) glycoprotein of SARS-CoV-2 in vascular leakage is described. Using a combination of in vitro TEER assays to assess cell junction integrity and in vivo extravasation tests (primarily dermal leak) the authors lay out a provocative series of experiments implicating integrins, TGF-beta, ECM remodeling enzymes, and glycosaminoglycans in S-mediated barrier dysfunction. Although the model systems used in this study are highly artificial, these results – if substantiated in additional systems – could provide important clues regarding COVID-19 pathogenesis. Studying complex multi-organ disease processes like this is very difficult, and the authors deserve praise for attempting to bring mechanistic insights to this significant yet understudied problem.

MAJOR COMMENTS

The authors could do a much better job articulating how they think all of these host factors are interrelated. As it currently stands, the manuscript appears to bounce from one host factor to the next, largely based upon prior evidence from Dengue NS1. Clearly stating (and returning to) a working hypothesis would help the reader keep up.

Many of the experiments have groups with $n=2$, and (based on the description in the figure legend) were not repeated. This does not meet the quality/reproducibility standards of a journal such as Nature Communications.

Most importantly, the experimental design could be improved to convince the reader that the models of S-mediated vascular leakage developed in this study are specific to SARS-CoV-2. For example, instead of using PBS or DENV NS1 as comparators, it would be extremely compelling to see some of these experiments repeated with the Spike protein of another beta-coronavirus (e.g., OC43) and an alpha-coronavirus (e.g., NL63). Though these other coronaviruses have S proteins that are related to that of SARS-CoV-2, both OC43 and NL63 are seasonal coronaviruses that clearly do not cause vascular leakage. Such a comparison would greatly increase the reader's confidence that the systems employed correlate with pathophysiology in humans.

MINOR COMMENTS

In all of the graphs, it would be good to see the individual datapoints.

Line 39 - SARS-CoV-2 is no longer "emerging"... I would say it has fully emerged.

Line 40 - mention that Coronaviridae is +ssRNA virus family

Line 44 - endothelial and epithelial dysfunction are only part of the story, what about immune cell infiltration? Pneumocyte damage? This topic deserves much more nuance.

Lines 65-67 - NL63 does something similar. Although it is an alpha-corona, it also uses ACE2 as a receptor. Worth mentioning this.

Fig 1F - why is "untreated" condition not at 1.0?

Figure 1 - Many of these experiments are $n=2$. Were they repeated? It does not state that they were in the legend.

Figure 2 C,D - it would be good to see the splay (via error bars) in the untreated controls as well.

Figure 4 - again, $n=2$ without repeating is not publication-worthy. Were these experiments repeated? In an assay setup that is as complex as TEER, it is important to show that the system is

reproducible.

Line 319 - data should be shown, or this statement withdrawn

Response: We thank the reviewers for their thoughtful and constructive comments, which we have now implemented as highlighted below. We believe this has substantially improved our study.

REVIEWER COMMENTS

Reviewer #1 (Remarks to the Author):

The manuscript “SARS-CoV-2 triggers barrier dysfunction and vascular leak via integrins and TGF-B signaling” by Eva Harris and colleagues is an exceptional manuscript that investigated the mechanisms underlying SARS-CoV-2 Spike protein-induced hyperpermeability.

The manuscript is well written and provides both in vitro and in vivo assays. Authors have initially investigated different types of viral proteins and their effect on endothelial monolayers and later identified critical necessary pathways for Spike protein-induced endothelial dysfunction. The manuscript later discusses the relevance of the cross-talk between endothelial cells and ECM proteins in regards to SP pathogenicity and identify TGF-B as a critical signaler of injury.

The data is presented in a logic progression and the number of experiments is adequate. The reading is thus pleasant and of great interest due to the ongoing Pandemic.

Few minor comments:

- The first evidence of Spike protein-induced endothelial barrier dysfunction was provided by Prof. Ramirez et col. (<https://pubmed.ncbi.nlm.nih.gov/33053430/>). The manuscript would be improved by adding this important information.

Response: We have now added a reference to this manuscript and several others, including the one mentioned below, to our Discussion section, lines 409-418.

- In that manuscript and in a more recent one (<https://www.frontiersin.org/articles/10.3389/fphys.2022.812199/full>), SP was able to elicit barrier dysfunction with a the dose of 10-20 nM, while in this paper a much higher amount was used (ug/ml). Authors, should explain their decision to use such a high concentration of spike protein and discuss it within the text.

Response: We thank the Reviewer for bringing up this important point. We initially conducted dose-response experiments of Spike on HPMEC at concentrations in line with what has been detected circulating in COVID-19 patients or from amounts present in sputum of critically ill COVID-19 patients (extrapolated from reported viral loads), which range in the ng/mL-µg/mL levels but have been detected as high as 17.5 µg/ml (~97 nM).^{1,2,3,4} We also hypothesize that local concentrations of Spike accumulating in capillaries deep within tissues would be higher than levels circulating in patient bodily fluids. Once we determined the relevant concentrations for our phenotype, we utilized the concentration required for a consistent in vitro phenotype. Both manuscripts referenced by the Reviewer utilize concentrations of Spike (full-length Spike, S1, S2, and RBD domains) from 0.1-50 nM, with the majority of experiments conducted at ~20 nM. Our experiments are primarily conducted with full-length Spike at a concentration of 10 µg/ml (equivalent to ~50 nM), but we demonstrate activity as low as 2.5 µg/ml (~13 nM). Once converted to molarity, it becomes more apparent that the concentrations we use are overall comparable to the concentrations of Spike used within the other studies, albeit on the higher side. It is important to point out that our manuscript investigates an ACE2-independent phenotype for Spike, while both of the studies referenced investigate an ACE2-dependent phenotype. This is important to note, as we find that Spike interacts with ACE2-expressing cells

more readily than cells that do not express ACE2, introducing a critical difference in our experimental systems. Beyond these studies, others have reported a role for Spike in inflammatory responses or barrier function at a concentration of 10 µg/ml (~50 nM) or even higher at 100 nM (18 µg/ml).^{5,6,7,8} We have now modified our results section explaining why we utilized the concentrations of spike within this study and further discuss the potential differences between an ACE2-dependent vs. ACE2-independent pathway of S function.

Lines 351-357: "The levels of S used in our study ranged from 2.5-20 µg/mL, with the majority of experiments conducted at 10 µg/mL (equivalent to ~50 nM). These concentrations are in line with levels detected circulating in patients as well as extrapolated from viral loads detected in sputum from critically ill COVID-19 patients (ranging from ng/mL to µg/mL levels) (56, 60-64). We also hypothesize that local concentrations of S accumulating in capillaries deep within tissues would likely be higher than levels circulating in patient sera. Thus, the concentrations of S we utilized in our study are consistent with circulating levels found in severe COVID-19 patients."

Lines 409-418: "Further, the expression of ACE2 on the cell surface may influence the capacity of S proteins to interact with endothelial and epithelial cells and trigger barrier dysfunction. This may be the case for SARS-CoV-1 and HCoV-NL63, which both utilize ACE2 as an entry receptor (69, 70). The interaction of S from both SARS-CoV-1 and HCoV-NL63 with Vero-E6 cells has been shown to lead to downregulation of ACE2 expression, although via a different mechanism, which has been shown to contribute to tissue injury in the case of SARS-CoV-1 S (21, 23, 71). Importantly, several reports have demonstrated that SARS-CoV-2 S can also trigger inflammatory responses and perturb barrier function in an ACE2-dependent manner (24, 25, 27, 72, 73). It will be critical to understand the relative contribution of the ACE2-independent vs. ACE2-dependent pathways to vascular leak in vivo and define which pathways a given coronavirus S protein can trigger."

Reviewer #2 (Remarks to the Author):

The pathophysiology of clinical COVID-19 disease suffice to say is complicated. Many reports have indicated impacts of the innate immune response, inflammatory responses and the over-production and contributions of cytokines, including IL-6. In this report Biering et al demonstrate that components of the SARS-CoV-2 Spike protein can bind to GAGs and integrins on the cell surface and induce a TGF- β response resulting in barrier dysfunction and vascular leakage. This work builds upon the previous excellent studies done by this lab on the NS1 glycoprotein of dengue virus (DENV), a well known hemorrhagic virus. The approach and methodologies have been well established by this lab in demonstrating vascular leakage by the DENV NS1 and equally here they are performed to a high standard and the results are of a very conclusive and high standard. Albeit the focus is primarily on the Spike protein alone and not on virus infection itself. The results are of significance particularly with the administration of whole Spike and RBD components as vaccine antigens. Certainly the methodology is sound and the conclusions drawn from these results are appropriate. The biggest difficulty is the correlation between expression of the Spike proteins and the actual events happening during infection both in vitro and in vivo.

Major issues to amend:

1. The results need to be compared to infection. This can be done in both the HPMEC and HPMEC/ACE2 cells. One would expect that infection of HPMEC should induce the same response as protein alone in the absence of active infection and replication. This is also true in the Calu-3 cells where UV-inactivated virus should be used.

*Response: To determine if infectious SARS-CoV-2 can induce endothelial dysfunction in HPMEC, we inoculated cells with SARS-CoV-2 at a multiplicity of infection (MOI) of 5 or treated cells with SARS-CoV-2 Spike as a positive control. Twenty-four hours post-infection/treatment, cells were fixed and stained for cell surface levels of the endothelial glycocalyx component sialic acid. We found that levels of sialic acid in Spike-treated HPMEC and SARS-CoV-2-infected HPMEC were significantly lower compared to control cells (**new Figure S2E and S2F**). These data indicate that both recombinant SARS-CoV-2 Spike and full viral particles can trigger endothelial dysfunction in vitro. Further, the current Figure 1D demonstrates that vesicular stomatitis virus (VSV) particles pseudotyped with SARS-CoV-2 Spike are also sufficient to trigger endothelial hyperpermeability of HPMEC, providing additional evidence that virion-associated Spike can trigger barrier dysfunction (**Figure 1D**). We thank the reviewer for his/her suggestion to test UV-inactivated SARS-CoV-2 stocks as well, but we feel that our live virus and VSV-Spike experiments with HPMECs address the question raised.*

2. Some consideration should be made when comparing the relative dose of Spike vs NS1 vs VEGF. All different doses were used, so firstly why and is this related to molarity? What is the basis for using some much protein (ie. 10ug/mL) when it was suggested that only 250 ng/mL is required (line 311). Was 250ng/mL tried and what was the outcome?

*Response: We thank the Reviewer for bringing up this important point. As also explained in response to Reviewer #1, we initially conducted dose-response experiments of Spike on HPMEC at concentrations representative of what has been detected circulating in COVID-19 patients or from amounts present in sputum of critically ill COVID-19 patients (extrapolated from reported viral loads).^{1,2,3,4} We also hypothesize that local concentrations of Spike accumulating in capillaries deep within tissues would be higher than levels circulating in patient bodily fluids. Once we determined the relevant concentrations for our phenotype, we utilized the concentration required for a consistent in vitro phenotype. Our dose response experiment ranged from 250 ng/mL- 20 $\mu\text{g/mL}$, and all concentrations tested caused endothelial hyperpermeability in a dose-dependent manner (**Figure 1C**). We have now modified the Discussion section to explain why we utilized the concentrations of Spike we did in this study (see Lines 351-357). Concerning concentrations of dengue virus (DENV) NS1, 5 $\mu\text{g/mL}$ (the concentration utilized in Figure 1B) is representative of levels of circulating in patients with severe dengue, which range from ~1 to 10 $\mu\text{g/mL}$. Thus, the concentrations of both Spike and NS1 we utilized are representative of levels associated with severe cases of COVID-19 and dengue, respectively. Within our study, the levels of Spike used are equivalent to ~50 nM, while the amount of DENV NS1 is equivalent to ~100 nM. Finally, the 250 ng/mL referenced in the discussion section is actually a typo that was supposed to read 2500 ng/mL which we have now corrected. This value is derived from a study published by George et al., that investigated levels of Spike in the serum and urine of COVID-19 patients. This study found Spike circulating in the serum of some patients at 2,500 ng/mL even as high as 17,500 ng/mL. Taken together, the available literature indicates that the range of Spike detected in patients is in line with the levels we utilize in our study.¹*

Lines 351-357: "The levels of S used in our study ranged from 2.5-20 $\mu\text{g/mL}$, with the majority of experiments conducted at 10 $\mu\text{g/mL}$ (equivalent to ~50 nM). These concentrations are in line with levels detected circulating in patients as well as extrapolated from viral loads detected in sputum from critically ill COVID-19 patients (ranging from ng/mL to $\mu\text{g/mL}$ levels) (56, 60-64). We also hypothesize that local concentrations of S accumulating in capillaries deep within tissues would likely be higher than levels circulating in patient sera. Thus, the concentrations of S we utilized in our study are consistent with circulating levels found in severe COVID-19 patients."

3. Infection of mice should also be evaluated by histology for barrier dysfunction, this can be achieved with Evans Blue or similar

*Response: In response to the Reviewer's suggestion, barrier dysfunction was evaluated in mice using two different models. First, K18-hACE2 mice were infected with the WA/1 clinical isolate of SARS-CoV-2, and mice were sacrificed at the peak of disease (~7 days post-infection). Mouse lungs and small intestine sections were fixed in formalin overnight, then hematoxylin and eosin (H&E) staining was performed. We observed a significant influx of cellular infiltrate into the lungs and small intestine of SARS-CoV-2-infected mice compared to control mice. Further, we observed dissemination of red blood cells throughout lungs and small intestines in infected mice relative to control mice. Taken together, these observations are consistent with barrier dysfunction and bleeding within SARS-CoV-2-infected mice. These data have been included as new **Figure 3I-J** and **Figure S3E-F**.*

*To further examine SARS-CoV-2 vascular leak in vivo in a quantitative manner, we utilized our previously established 10kD-dextran-AF680 leak systemic leak model.⁹ In this experiment, we utilized C57BL/6J mice and infected them with a mouse-adapted strain of SARS-CoV-2 (MA10). At the peak of disease at 7 days post-infection, we administered a fluorescent tracer dye intravenously and then evaluated the levels of fluorescence in lungs of mice, finding a significant accumulation in infected mice relative to mock-infected mice, which positively correlated with the initial viral inoculum dose (**new Figure 3K and 3L**). To test for a role of integrins in SARS-CoV-2-mediated vascular leak, we utilized this systemic leak infection model and administered the integrin inhibitor ATN-161 to mice daily throughout the course of SARS-CoV-2 infection and again evaluated leak in infected mice at day 7 post-infection. We found, as before, that infection of mice triggered significant leak in the lungs relative to mock-infected conditions (**new Figure 6L and 6M**). Strikingly, while daily administration of ATN-161 had no effect on background levels of vascular leak in these mice, it did significantly inhibit SARS-CoV-2-triggered leak, suggesting that virus-induced vascular leak, comparably to Spike-triggered leak, requires integrins (**new Figure 6L and 6M**). In all, this new body of data not only suggests that SARS-CoV-2 infection triggers vascular leak in vivo, but also that the mechanism by which it does so requires integrins comparably to Spike treatment alone.*

4. The authors show that the RGD domain of Spike is sufficient to induce leakage, so then is this applicable to all other viral structural proteins that contain this motif? Importantly Adenovirus 5 that is used as a vaccine to deliver Spike. This is essence should induce dysfunction via both the virus vector and the encoded Spike protein.

Response: We agree that any protein containing an "RGD" motif would in theory trigger barrier dysfunction. Of course, this depends on the context of the "RGD" motif within a given protein, which would be influenced by the fold/structure of the protein dictating surface exposure and availability for binding integrins. This is also influenced by the concentrations of these RGD-containing proteins within tissues and their ability to interact with barrier cells (epithelial or endothelial). That said, while this predicts that both the Ad5 vector and vaccine-vectored Spike may be able to trigger some sort of transient barrier dysfunction, our work suggests that the concentrations present post-vaccination as well as the intramuscular administration route would not be sufficient to cause leak. What is important to consider here is that Spike activates release of the growth factor TGF- β , and while this may contribute to disease severity in the context of SARS-CoV-2 infection, it has plenty of critical roles in development and in the immune system that are protective or simply non-pathogenic on its own.

5. A report published by Rauti et al (2021) eLife expressed all of the SARS-CoV-2 proteins in

HUVECs and performed similar assays to those described here. However, Spike was not observed to induce leakage in their studies, many of the other non-structural proteins did. How can such differences be explained? Did the authors also try HUVECs? Although the point below may counter this

Response: The key difference between Rauti et al. and our study is that Rauti et al. utilize lentivirus transduction to overexpress SARS-CoV-2 proteins within HUVECs, while our study uses recombinant proteins and viral particle-associated Spike added exogenously to HPMECs, which we believe is more representative of how the majority of cells would encounter the Spike protein during infection. Further, Rauti et al. don't seem to have a protein expression control experiment to show that their lentivirus transduction was successful in inducing detectable overexpression of Spike. From our experience, Spike is often expressed at low levels following lentivirus transduction. Thus, it is hard to conclude if they achieved a high enough expression of Spike protein to trigger barrier dysfunction. That said, and as this Reviewer points out, others have demonstrated that HUVECs are sensitive to Spike-mediated barrier dysfunction. In fact, we are also able to show that Spike triggers barrier dysfunction in both HBMEC (brain) and HUVEC (umbilical vein) in addition to HPMEC (lung). In this study, we are focusing on the capacity of Spike to trigger vascular leak in the lung (thus our focus on lung endothelial and epithelial cells), but our future studies will expand on the function of Spike in interacting with diverse tissue-specific endothelial cell lines.

6. Would also comment on the recent publication by Robles et al (2022) JBC who have shown that Spike, RBD and RGD induce hyperpermeability in HUVECs via $\alpha 5\beta 1$ integrin that can be restricted by treatment with Volciximab and ATN-161 so some novelty is lost in this report

Response: We have now modified the discussion section of our manuscript to include discussion of this study (Lines 378-383). We are encouraged that other studies have verified the findings within our study, but we would also like to highlight that our study defines multiple distinct steps in the mechanism by which Spike triggers barrier dysfunction, including glycan binding, integrin activation, and finally TGF- β production, all of which seem to be critical for this pathway. Further, we also extend these critical findings into SARS-CoV-2 infection models. Thus, while we agree that some similar observations are reported in Robles et al., there is still a large body of novel work being presented in our study.

Reviewer #3 (Remarks to the Author):

In “SARS-CoV-2 Spike triggers barrier dysfunction and vascular leak via integrins and TGF- β signaling,” Biering et al. describe a study in which a novel role for the Spike (S) glycoprotein of SARS-CoV-2 in vascular leakage is described. Using a combination of in vitro TEER assays to assess cell junction integrity and in vivo extravasation tests (primarily dermal leak) the authors lay out a provocative series of experiments implicating integrins, TGF-beta, ECM remodeling enzymes, and glycosaminoglycans in S-mediated barrier dysfunction. Although the model systems used in this study are highly artificial, these results – if substantiated in additional systems – could provide important clues regarding COVID-19 pathogenesis. Studying complex multi-organ disease processes like this is very difficult, and the authors deserve praise for attempting to bring mechanistic insights to this significant yet understudied problem.

MAJOR COMMENTS

The authors could do a much better job articulating how they think all of these host factors are interrelated. As it currently stands, the manuscript appears to bounce from one host factor to the next, largely based upon prior evidence from Dengue NS1. Clearly stating (and returning to) a working hypothesis would help the reader keep up.

Response: We appreciate this feedback from the Reviewer. We lay out our study as a step-wise mechanistic investigation of if/how SARS-CoV-2 Spike causes barrier dysfunction and vascular leak in vitro and in vivo. We agree that we introduce many host factors in this study and thus have created a diagram as Figure 7K that highlights the central hypothesis of how all of these host factors are interrelated. We have also worked hard to introduce our reasoning for why we hypothesized these factors to be involved in S-mediated barrier dysfunction and are highlighting some instances below. Lines 139-144 lay out the logic of why we investigated whether S disrupts the EGL. On lines 204-210, we explain why we anticipate sulfated glycans to be important for initial binding of S to barrier cells. Lines 227-241 highlight enzymes that are ubiquitously known to be involved in ECM reorganization. On lines 244-249, we discuss the need to identify additional key players in this pathway; this RNA-Seq analysis identifies integrins and TGF- β to be potential key players. On lines 271-286, we explain why integrins may be involved in this pathway and how they regulate barrier function. Lines 317-321 explain the connection between integrins and TGF- β signaling. Finally, paragraph 3 (Lines 367-382) of the Discussion further returns to our central hypothesis and describes how we think all of these host factors coordinate S-mediated barrier dysfunction.

Many of the experiments have groups with n=2, and (based on the description in the figure legend) were not repeated. This does not meet the quality/reproducibility standards of a journal such as Nature Communications.

Response: We have now repeated experiments such that experiments have at least an n=3.

Most importantly, the experimental design could be improved to convince the reader that the models of S-mediated vascular leakage developed in this study are specific to SARS-CoV-2. For example, instead of using PBS or DENV NS1 as comparators, it would be extremely compelling to see some of these experiments repeated with the Spike protein of another beta-coronavirus (e.g., OC43) and an alpha-coronavirus (e.g., NL63). Though these other coronaviruses have S proteins that are related to that of SARS-CoV-2, both OC43 and NL63 are seasonal coronaviruses that clearly do not cause vascular leakage. Such a comparison would greatly increase the reader's confidence that the systems employed correlate with pathophysiology in humans.

*Response: To determine the impact of Spike proteins from other coronaviruses, we utilized recombinant Spike (both full-length and receptor binding domain [RBD]) from human coronavirus 229E (HCoV-229E) and HCoV-OC43 and tested their capacity to trigger endothelial hyperpermeability via a transendothelial electrical resistance (TEER) assay. Intriguingly, while SARS-CoV-2 Spike (both full-length and RBD) triggered endothelial hyperpermeability of HPMEC, Spike proteins and RBDs from both HCoV-229E and HCoV-OC43 failed to do so, suggesting that SARS-CoV-2 S/RBD contain specific motifs capable of triggering endothelial hyperpermeability of HPMEC (**New Fig. S1H**). We further discuss the implications of these new findings in the Discussion highlighted below.*

Lines 404-410: "Our observation that S from SARS-CoV-2 but not from HCoV-229E or HCoV-OC43 triggers endothelial hyperpermeability of HPMECs suggests that the capacity to trigger barrier dysfunction in these lung cells is not conserved equivalently among all

coronaviruses. We hypothesize that the increased capacity of SARS-CoV-2 S to interact with heparan sulfate and integrins on HPMEC may explain this specificity, but additional studies are required to test this possibility (18, 20). Further, the expression of ACE2 on the cell surface may influence the capacity of S proteins to interact with endothelial and epithelial cells and trigger barrier dysfunction.”

MINOR COMMENTS

In all of the graphs, it would be good to see the individual datapoints.

Response: We have now revised all graphs to show individual datapoints, as requested.

Line 39 - SARS-CoV-2 is no longer “emerging”... I would say it has fully emerged.

Response: We have removed the word “emerging”.

Line 40 - mention that Coronaviridae is +ssRNA virus family

Response: This detail is currently included on line 52.

Line 44 - endothelial and epithelial dysfunction are only part of the story, what about immune cell infiltration? Pneumocyte damage? This topic deserves much more nuance.

Response: We have now expanded our Discussion section to discuss that COVID-19 disease pathology is complex and that vascular leak/ barrier dysfunction is only one component.

Lines 447-451: “It is also important to consider that reported COVID-19 disease manifestations are diverse and may be explained by factors other than vascular leak, including pneumocyte damage resulting from immune cell infiltration and viral infection. Understanding the relative contribution of vascular leak to COVID-19 disease severity will undoubtedly be complicated but is nevertheless a critical question.”

Lines 65-67 - NL63 does something similar. Although it is an alpha-corona, it also uses ACE2 as a receptor. Worth mentioning this.

Response: We have now expanded our discussion section to compare and contrast the mechanism by which SARS-CoV-1 Spike and HCoV-NL63 Spike interact with and downregulate ACE2 and how this influences lung injury through ACE2 ectodomain shedding (Lines 409-418).

Lines 409-418 “Further, the expression of ACE2 on the cell surface may influence the capacity of S proteins to interact with endothelial and epithelial cells and trigger barrier dysfunction. This may be the case for SARS-CoV-1 and HCoV-NL63, which both utilize ACE2 as an entry receptor (69, 70). The interaction of S from both SARS-CoV-1 and HCoV-NL63 with Vero-E6 cells has been shown to lead to downregulation of ACE2 expression, although via a different mechanism, which has been shown to contribute to tissue injury in the case of SARS-CoV-1 S (21, 23, 71). Importantly, several reports have demonstrated that SARS-CoV-2 S can also trigger inflammatory responses and perturb barrier function in an ACE2-dependent manner (24, 25, 27, 72, 73). It will be critical to understand the relative contribution of the ACE2-independent vs. ACE2-dependent pathways to vascular leak in vivo and define which pathways a given coronavirus S protein can trigger.”

Fig 1F - why is “untreated” condition not at 1.0?

Response: We thank the reviewer for pointing this out, it was indeed a mistake on our part that we have now corrected.

Figure 1 - Many of these experiments are n=2. Were they repeated? It does not state that they were in the legend.

Response: We have now repeated experiments in order to have at least an n of 3.

Figure 2 C,D - it would be good to see the splay (via error bars) in the untreated controls as well.

Response: We are including Figure 2C and 2D with all conditions, including untreated controls, with raw MFI graphed (as Reviewer Figure 1) to demonstrate the spread and reproducibility of our data. Because we are comparing multiple different components within the same graph, we have normalized these data in the figures in the manuscript so they can be better compared and visualized together.

Reviewer Figure 1: Immunofluorescence assay (IFA) with HPMEC treated with 10 µg/mL of SARS-CoV-2 Spike or left untreated. 24 hours post treatment cells were fixed and stained for the indicated components. Raw mean fluorescence intensity (MFI) is displayed and graphed as mean +/- SEM. These graphs are the unnormalized data from Figures 2C and 2D.

Figure 4 - again, n=2 without repeating is not publication-worthy. Were these experiments repeated? In an assay setup that is as complex at TEER, it is important to show that the system is reproducible.

Response: We have now repeated experiments such that n=3.

Line 319 - data should be shown, or this statement withdrawn

Response: We have now withdrawn this statement.

References

- (1) George, S.; Pal, A. C.; Gagnon, J.; Timalisina, S.; Singh, P.; Vydyam, P.; Munshi, M.; Chiu, J. E.; Renard, I.; Harden, C. A.; et al. Evidence for SARS-CoV-2 Spike Protein in the Urine of COVID-19 Patients. *Kidney360* **2021**, *2* (6), 924. DOI: 10.34067/KID.0002172021.
- (2) Avolio, E.; Carrabba, M.; Milligan, R.; Kavanagh Williamson, M.; Beltrami, A.; Gupta, K.; Elvers, K. T.; Gamez, M.; Foster, R.; Gillespie, K.; et al. The SARS-CoV-2 Spike protein disrupts human cardiac pericytes function through CD147-receptor-mediated signalling: a potential non-infective mechanism of COVID-19 microvascular disease. *Clin Sci (Lond)* **2021**. DOI: 10.1042/CS20210735.
- (3) Pan, Y.; Zhang, D.; Yang, P.; Poon, L. L. M.; Wang, Q. Viral load of SARS-CoV-2 in clinical samples. *Lancet Infect Dis* **2020**, *20* (4), 411-412. DOI: 10.1016/S1473-3099(20)30113-4.
- (4) Bar-On, Y. M.; Flamholz, A.; Phillips, R.; Milo, R. SARS-CoV-2 (COVID-19) by the numbers. *Elife* **2020**, *9*. DOI: 10.7554/eLife.57309.
- (5) Raghavan, S.; Kenchappa, D. B.; Leo, M. D. SARS-CoV-2 Spike Protein Induces Degradation of Junctional Proteins That Maintain Endothelial Barrier Integrity. *Front Cardiovasc Med* **2021**, *8*, 687783. DOI: 10.3389/fcvm.2021.687783.
- (6) Robles, J. P.; Zamora, M.; Adan-Castro, E.; Siqueiros-Marquez, L.; Martinez de la Escalera, G.; Clapp, C. The spike protein of SARS-CoV-2 induces endothelial inflammation through integrin $\alpha 5\beta 1$ and NF- κB signaling. *J Biol Chem* **2022**, *298* (3), 101695. DOI: 10.1016/j.jbc.2022.101695.
- (7) Moutal, A.; Martin, L. F.; Boinon, L.; Gomez, K.; Ran, D.; Zhou, Y.; Stratton, H. J.; Cai, S.; Luo, S.; Gonzalez, K. B.; et al. SARS-CoV-2 spike protein co-opts VEGF-A/neuropilin-1 receptor signaling to induce analgesia. *Pain* **2021**, *162* (1), 243-252. DOI: 10.1097/j.pain.0000000000002097.
- (8) Colunga Biancatelli, R. M. L.; Solopov, P. A.; Sharlow, E. R.; Lazo, J. S.; Marik, P. E.; Catravas, J. D. The SARS-CoV-2 spike protein subunit S1 induces COVID-19-like acute lung injury in K18-hACE2 transgenic mice and barrier dysfunction in human endothelial cells. *American Journal of Physiology-Lung Cellular and Molecular Physiology* **2021**, *321* (2), L477-L484. DOI: 10.1152/ajplung.00223.2021 (accessed 2021/09/14).
- (9) Puerta-Guardo, H.; Glasner, D. R.; Espinosa, D. A.; Biering, S. B.; Patana, M.; Ratnasiri, K.; Wang, C.; Beatty, P. R.; Harris, E. Flavivirus NS1 Triggers Tissue-Specific Vascular Endothelial Dysfunction Reflecting Disease Tropism. *Cell Rep* **2019**, *26* (6), 1598-1613.e1598. DOI: 10.1016/j.celrep.2019.01.036.

Reviewer comments, second round –

Reviewer #1 (Remarks to the Author):

No more comments.

Reviewer #2 (Remarks to the Author):

All of my comments have been addressed satisfactorily in this revised manuscript

Reviewer #3 (Remarks to the Author):

The reviewers have taken our collective comments/suggestions seriously, and I believe the manuscript is greatly strengthened and worthy of publication in Nature Communications. My only comment/concern relates to normalization of data... for example, in "Reviewer Figure 1," the MFI values are shown and the untreated controls have a relatively large degree of splay; however, in the corresponding figures in the manuscript (2G,H) which display normalized data, all untreated controls are at 100%. This implies that each data point was treated independently and not as a replicate... please consult a statistician on whether or not this is appropriate.

Response to Reviewers

Response: We thank the reviewers for their support of our study and appreciate the current and previous comments which we have implemented which we believe have strengthened our study.

Reviewer #1 (Remarks to the Author):

No more comments.

Reviewer #2 (Remarks to the Author):

All of my comments have been addressed satisfactorily in this revised manuscript

Reviewer #3 (Remarks to the Author):

The reviewers have taken our collective comments/suggestions seriously, and I believe the manuscript is greatly strengthened and worthy of publication in Nature Communications. My only comment/concern relates to normalization of data... for example, in "Reviewer Figure 1," the MFI values are shown and the untreated controls have a relatively large degree of splay; however, in the corresponding figures in the manuscript (2G,H) which display normalized data, all untreated controls are at 100%. This implies that each data point was treated independently and not as a replicate... please consult a statistician on whether or not this is appropriate.

Response: We thank this reviewer for their support of our study and we appreciate this comment. We have consulted with a statistician who has confirmed that the data normalization in figures 2G, 2H, and beyond are correct and appropriate given the fact that these data are derived from distinct biological replicates.